# On Single Index Models beyond Gaussian Data

**Joan Bruna**
New York University

**Loucas Pillaud-Vivien**
New York University
Flatiron Institute

**Aaron Zweig**
New York University

## Abstract

Sparse high-dimensional functions have arisen as a rich framework to study the behavior of gradient-descent methods using shallow neural networks, showcasing their ability to perform feature learning beyond linear models. Amongst those functions, the simplest are single-index models $f(x) = \phi(x \cdot \theta^*)$, where the labels are generated by an arbitrary non-linear scalar link function $\phi$ applied to an unknown one-dimensional projection $\theta^*$ of the input data. By focusing on Gaussian data, several recent works have built a remarkable picture, where the so-called information exponent (related to the regularity of the link function) controls the required sample complexity. In essence, these tools exploit the stability and spherical symmetry of Gaussian distributions. In this work, building from the framework of Ben Arous et al. [2021], we explore extensions of this picture beyond the Gaussian setting, where both stability or symmetry might be violated. Focusing on the planted setting where $\phi$ is known, our main results establish that Stochastic Gradient Descent can efficiently recover the unknown direction $\theta^*$ in the high-dimensional regime, under assumptions that extend previous works Yehudai and Shamir [2020], Wu [2022].

## 1 Introduction

Over the past years, there has been sustained effort to enlarge our mathematical understanding of high-dimensional learning, particularly when using neural networks trained with gradient-descent methods — highlighting the interplay between algorithmic, statistical and approximation questions. An essential, distinctive aspect of such models is their ability to perform *feature learning*, or to extract useful low-dimensional features out of high-dimensional observations.

An appealing framework to rigorously analyze this behavior are sparse functions of the form $f(x) = \phi(\Theta_*^\top x)$, where the labels are generated by a generic non-linear, low-dimensional function $\phi : \mathbb{R}^k \to \mathbb{R}$ of linear features $\Theta_*^\top x$, with $\Theta_* \in \mathbb{R}^{d \times k}$ with $k \ll d$. While the statistical and approximation aspects of such function classes are by now well-understood Barron [1993], Bach [2017], the outstanding challenge remains computational, in particular in understanding the ability of gradient-descent methods to succeed. Even in the simplest setting of *single-index models* ($k = 1$), and assuming that $\phi$ is known, the success of gradient-based learning depends on an intricate interaction between the data distribution $x \sim \nu$ and the 'link' function $\phi$; and in fact computational lower bounds are known for certain such choices Yehudai and Shamir [2020], Song et al. [2021], Goel et al. [2020], Diakonikolas et al. [2017], Shamir [2018].

Positive results thus require to make specific assumptions, either about the data, or about the link function, or both. On one end, there is a long literature, starting at least with Kalai and Sastry [2009], Shalev-Shwartz et al. [2010], Kakade et al. [2011], that exploits certain properties of $\phi$, such as invertibility or monotonicity, under generic data distributions satisfying mild anti-concentration properties Soltanolkotabi [2017], Frei et al. [2020], Yehudai and Shamir [2020], Wu [2022]. On the other end, by focusing on canonical high-dimensional measures such as the Gaussian distribution, the seminal works Ben Arous et al. [2021], Dudeja and Hsu [2018] built a harmonic analysis

37th Conference on Neural Information Processing Systems (NeurIPS 2023).

framework of SGD, resulting in a fairly complete picture of the sample complexity required to learn generic link functions $\phi$, and revealing a rich asymptotic landscape beyond the proportional regime $n \asymp d$, characterized by the number of vanishing moments, or *information exponent* $s$ of $\phi$, whereby $n \asymp d^{s-1}$ samples are needed for recovery. Since then, several authors have built and enriched this setting to multi-index models Abbe et al. [2022, 2023], Damian et al. [2022], Arnaboldi et al. [2023], addressing the semi-parametric learning of the link function Bietti et al. [2022], as well as exploring SGD-variants Ben Arous et al. [2022], Barak et al. [2022], Berthier et al. [2023], Chen et al. [2023]. This harmonic analysis framework relies on two key properties of the Gaussian measure and their interplay with SGD: its spherical symmetry and its stability by linear projection. Together, they provide an optimization landscape that is well-behaved in the limit of infinite data, and enable SGD to escape the 'mediocrity' of initialisation, where the initial direction $\theta_0$, in the high-dimensional setting, has vanishingly small correlation $|\theta_0 \cdot \theta^*| \simeq 1/\sqrt{d}$ with the planted direction $\theta^*$.

In this work, we study to what extent the 'Gaussian picture' is robust to perturbations, focusing on the planted setting where $\phi$ is known. Our motivation comes from the fact that real data is rarely Gaussian, yet amenable to being approximately Gaussian via CLT-type arguments. We establish novel positive results along two main directions: (i) when spherical symmetry is preserved but stability is lost, and (ii) when spherical symmetry is lost altogether. In the former, we show that spherical harmonics can be leveraged to provide a benign optimization landscape for SGD under mild regularity assumptions, for initialisations that can be reached with constant probability with the same sample complexity as in the Gaussian case. In the latter, we quantify the lack of symmetry with robust projected Wasserstein distances, and show that for 'quasi-symmetric' measures with small distance to the Gaussian reference, SGD efficiently succeeds for link functions with information exponent $s \leq 2$. Finally, using Stein's method, we address substantially 'non-symmetric' distributions, demonstrating the strength and versatility of the harmonic analysis framework.

## 2    Preliminaries and Problem Setup

The focus of this work is to understand regression problems with input/output data $(x, y) \in \mathbb{R}^d \times \mathbb{R}$ generated by *single-index* models. This is a class of problems where the data labels are produced by a non-linear map of a one-dimensional projection of the input, that is

$$y = \phi(x \cdot \theta^*), \tag{1}$$

where $\phi : \mathbb{R} \to \mathbb{R}$ is also known as the *link function*, and $\theta^* \in \mathcal{S}_{d-1}$, the sphere of $\mathbb{R}^d$, is the *hidden direction* that the models wants to learn. Quite naturally, the learning is made through the family of generalized linear predictors $\mathcal{H} = \{\phi_\theta : x \to \phi(x \cdot \theta), \text{ for } \theta \in \mathcal{S}_{d-1}\}$, built upon the link function (which is assumed known) and parametrized by the sphere.

**Loss function.**    We assume that the input data is distributed according to a probability $\nu \in \mathcal{P}(\mathbb{R}^d)$. Equation (1) implies that the target function that produces the labels, $\phi_{\theta^*}$, lies in this parametric class. The overall loss classically corresponds to the average over all the data of the square penalisation $l(\theta, x) := (\phi_\theta(x) - \phi_{\theta^*}(x))^2$ so that the population loss writes

$$L(\theta) := \mathbb{E}_\nu \left[ \left( \phi(x \cdot \theta) - \phi(x \cdot \theta^*) \right)^2 \right] = \| \phi_\theta - \phi_{\theta^*} \|_{L^2_\nu}^2, \tag{2}$$

where we used the notation $\|f\|_{L^p_\nu}^p = \mathbb{E}_\nu[|f|^p]$, valid for all $p \in \mathbb{N}^*$. Let us put emphasis on the fact that the loss $L$ is a non-convex function of the parameter $\theta$, hence it is not *a priori* guaranteed that gradient-based method are able to retrieve the ground-truth $\theta^*$. This often requires a precise analysis of the *loss landscape*, and where the high-dimensionality can play a role of paramount importance: we place ourselves in this high-dimensional setting for which the dimension is fixed but considered very large $d \gg 1$. Finally, we assume throughout the article that $\phi_\theta$ belongs to the weighted Sobolev space $W^{1,4}_\nu := \{\phi, \text{ such that } \sup_{\theta \in \mathcal{S}_{d-1}} \left[ \|\phi_\theta\|_{L^4_\nu} + \|\phi'_\theta\|_{L^4_\nu} \right] < \infty\}$.

**Stochastic gradient descent.**    To recover the signal given by $\theta^* \in \mathcal{S}_{d-1}$, we run *online stochastic gradient descent* (SGD) on the sphere $\mathcal{S}_{d-1}$. This corresponds to having at each iteration $t \in \mathbb{N}^*$ a *fresh sample* $x_t$ drawn from $\nu$ and independent of the filtration $\mathcal{F}_t = \sigma(x_1, \ldots, x_{t-1})$ and performing a spherical gradient step, with step-size $\delta > 0$, with respect to $\theta \to l(\theta, x_t)$:

$$\theta_{t+1} = \frac{\theta_t - \delta \nabla_\theta^{\mathcal{S}} l(\theta_t, x_t)}{\left| \theta_t - \delta \nabla_\theta^{\mathcal{S}} l(\theta_t, x_t) \right|}, \qquad \text{with initialization } \theta_0 \sim \text{Unif}(\mathcal{S}_{d-1}), \tag{3}$$

An important scalar function that enables to track the progress of the SGD iterates is the correlation with the signal $m_\theta := \theta \cdot \theta^* \in [-1,1]$. We will drop the subscript in case there is no ambiguity. Note that, due to the high-dimensionality of the setup, we have the following lemma:

**Lemma 2.1.** *For all $a > 0$, we have $\mathbb{P}_{\theta_0}(m_{\theta_0} \geq a/\sqrt{d}) \leq a^{-1}e^{-a^2/4}$. Additionally, for any $\delta > 0$ such that $\max\{a, \delta\} \leq \sqrt{d}/4$, we have the lower bound: $\mathbb{P}_{\theta_0}(m_{\theta_0} \geq a/\sqrt{d}) \geq \frac{\delta}{4}e^{-(a+\delta)^2}$.*

This fact implies that, when running the algorithm in practice, it is initialized with high probability near the equator of $\mathcal{S}_{d-1}$, or at least in a band of typical size $1/\sqrt{d}$ (see Figure 1 for a schematic illustration of this fact). Finally, we use the notation $\nabla_\theta^{\mathcal{S}}$ to denote the spherical gradient, that is $\nabla_\theta^{\mathcal{S}} l(\theta, x) = \nabla_\theta l(\theta, x) - (\nabla_\theta l(\theta, x) \cdot \theta)\theta$. As $\nabla_\theta^{\mathcal{S}} l(\cdot, x_t)$ is an unbiased estimate of $\nabla_\theta^{\mathcal{S}} L$, it is expected that the latter gradient field rules how the SGD iterates travel across the loss landscape.

**Loss landscape in the Gaussian case.** As stressed in the introduction, this set-up has been studied by Dudeja and Hsu [2018], Ben Arous et al. [2021] in the case where $\nu$ is the standard Gaussian, noted as $\gamma$ here to avoid any confusion for later. Let us comment a bit this case to understand what can be the typical landscape of this single-index problem. Thanks to the spherical symmetry, the loss admits a scalar summary statistic, given precisely by the correlation $m_\theta$. Moreover, the loss admits an explicit representation in terms of the Hermite decomposition of the link function $\phi$: if $\{h_j\}_j$ denotes the orthonormal basis of Hermite polynomials of $L_\gamma^2$, then $L(\theta) = 2 \sum_j |\langle \phi, h_j \rangle|^2 (1 - m^j) := \bar{\ell}(m)$. As a result, the gradient field projected along the signal is a (locally simple) *positive function* of the correlation that behaves similarly to

$$-\nabla_\theta^{\mathcal{S}} L(\theta) \cdot \theta^* \simeq Cm^{s-1}(1-m), \tag{4}$$

where $s \in \mathbb{N}^*$ is the index of the first non-zero of the Hermite coefficients $\{\langle \phi, h_j \rangle\}_j$. This has at least three important consequences for the gradient flow: (i) if initialized positively, the correlation is an increasing function along the dynamics and there is no bad local minima in the loss landscape, (ii) the parameter $s \in \mathbb{N}^*$ controls the flatness of the loss landscape near the origin and therefore controls the optimization speed of SGD in this region (iii) as soon as $m$ is large enough, the contractive term $1 - m$ makes the dynamics converge exponentially fast.

**Loss landscape in general cases.** Obviously for general distributions $\nu$, the calculation presented in Eq.(4) is no-longer valid. However, the crux of the present paper is that properties (i)-(ii)-(iii) are robust to the change of distribution and can be shown to be preserved under small adaptations. More precisely, we have the following definition.

**Definition 2.2** (Local Polynomial Growth)**.** *We say that $L$ has the local polynomial growth of order $k \in \mathbb{N}^*$ and scale $b \geq 0$, if there exists $C > 0$ such that for all $m_\theta \geq b$,*

$$-\nabla_\theta^{\mathcal{S}} L(\theta) \cdot \theta^* \geq C(1 - m_\theta)(m_\theta - b)^{k-1} . \tag{5}$$

*In such a case we say that $L$ satisfies* $\mathsf{LPG}(k, b)$.

In this definition, and as showed later in specific examples given in Section 4, the scale parameter $b$ should be thought as a small parameter proportional to $1/\sqrt{d}$. If $\nu$ is Gaussian, we can rewrite Eq.(4) and show that $L$ verifies $\mathsf{LPG}(s, 0)$ for $s \in \mathbb{N}^*$, referred to as the *information exponent* of the problem in Ben Arous et al. [2021]. An important consequence of satisfying $\mathsf{LPG}(k, b)$ is that the the population landscape is free of bad local minima outside the equatorial band $\Sigma_b := \{\theta \in \mathcal{S}_{d-1} , m_\theta \leq b\}$. Therefore, when $b$ is of scale $1/\sqrt{d}$, Lemma 2.1 indicates that one can efficiently produce initializations that avoid it. Hence, this property is the fundamental ingredient that enables the description of the path taken by SGD that we derive it in the next Section. Section 4 is devoted to showcasing generic examples when this property is satisfied.

## 3 Stochastic Gradient Descent under $\mathsf{LPG}$

In this section, we derive the main results on the trajectory of the stochastic gradient descent. They state that the property $\mathsf{LPG}(s, b/\sqrt{d})$ is in fact *sufficient* to recover the same quantitative guarantees as the one depicted in Ben Arous et al. [2021], despite the lack of Gaussianity of the distribution $\nu$. Recall that the recursion satisfied by the SGD iterates is given by Eq.(3). To describe their movement, let us introduce the following notations: for all $t \in \mathbb{N}^*$, we denote the normalization by $r_t := |\theta_t - \delta \nabla_\theta^{\mathcal{S}} \ell(\theta_t, x_t)|$ and the martingale induced by the stochastic gradient descent as $M_t := l(\theta_t, x_t) - \mathbb{E}_\nu[l(\theta_t, x)]$.

**Moment growth assumptions.** To be able to analyse the SGD dynamics, we make the following assumptions on the moments of the martingale increments induced by the random sampling. Let us denote for all $\theta \in \mathbb{R}^d$, $x_\theta = x \cdot \theta \in \mathbb{R}$ and $C(u,v) = \phi'(u)\phi(v)$, for $u, v \in \mathbb{R}$.

**Assumption 3.1** (Moment Growth Assumption). *There exists a constant $K > 0$, independent of the dimension $d$, such that*

$$\sup_{\theta \in \mathcal{S}_{d-1}} \mathbb{E}_x \left[ x_{\theta_*}^2 C^2(x_\theta, x_{\theta_*}) \right] \vee \mathbb{E}_x \left[ x_\theta^2 C^2(x_\theta, x_{\theta_*}) \right] \leq K, \qquad \text{and}, \qquad (6)$$

$$\sup_{\theta \in \mathcal{S}_{d-1}} \mathbb{E}_x \left[ |x|^{2k} C^2(x_\theta, x_{\theta_*}) \right] \leq K d^k, \qquad \text{for } k = 1, 2. \qquad (7)$$

A precise care is given to the dependency in the dimension in the upper bound to match the practical cases that we later discuss in Section 4. Note that these assumptions are typically true for sub-gaussian random variables if $\phi$ belongs to a Sobolev regularity class. These assumptions are similar to the one given in Eqs. (1.3)-(1.4) in Arous et al. [2021] for the Gaussian case. In all the remainder of the section we assume that Assumption 3.1 is satisfied.

**Tracking the correlation.** Recall that the relevant signature of the dynamics is the one-dimensional correlation $m_t = \theta_t \cdot \theta^*$. For infinitesimal step-sizes $\delta \to 0$, it is expected that $\theta_t$ follow the spherical gradient flow $\dot{\theta}_t = -\nabla_\theta^{\mathcal{S}} L(\theta_t)$, that translates to the evolution on the summary statistics $m_t$ whereby

$$\dot{m}_t = -\nabla_\theta^{\mathcal{S}} L(\theta_t) \cdot \theta^*. \qquad (8)$$

The main idea behind the result of this section is to show that, even if the energy landscape near $m = m_0$ is rough at scale $1/\sqrt{d}$, the noise induced by SGD does not prevent $m$ to *grow as the idealized dynamics described by the ODE* (8). Let us write the iterative recursion followed by $(m_t)_{t \geq 0}$: with the notation recalled above, for $t \in \mathbb{N}^*$, we have

$$m_{t+1} = \frac{1}{r_t} \left( m_t - \delta \nabla_\theta^{\mathcal{S}} L(\theta_t) \cdot \theta^* - \delta \nabla_\theta^{\mathcal{S}} M_t \cdot \theta^* \right). \qquad (9)$$

With this dynamics at hand, the proof consists in controlling both the discretization error through $r_t$ and the directional martingale induced by the term $\delta \nabla_\theta^{\mathcal{S}} M_t \cdot \theta^*$.

**Weak recovery.** As it is the case for the gradient flow, most of the time spent by the SGD dynamics is near the equator, or more precisely in a band of the type $\Sigma_{b,c} = \{\theta \in \mathcal{S}_{d-1}, b/\sqrt{d} \leq m_\theta \leq c/\sqrt{d}\}$, where $b < c$ are constants independent of the dimension. Hence, the real first step of the dynamics is to go out any of these bands. This is the reason why it is natural to define $S_a := \{\theta \in \mathcal{S}_{d-1}, m_\theta \geq a\}$, the spherical cap of level $a \in (0,1)$ as well as the hitting time

$$\tau_a^+ := \inf\{t \geq 0, m_{\theta_t} \geq a\}, \qquad (10)$$

which corresponds to the first time $(\theta_t)_{t \geq 0}$ enters in $S_a$. We fix a dimension-independent constant $a = 1/2$, and refer to the related hitting time $\tau_{1/2}^+$ as the *weak recovery time* of the algorithm.

**Theorem 3.2** (Weak Recovery). *Let $(\theta_t)_{t \geq 0}$ follow the SGD dynamics of Eq.(3) and let $L$ satisfy* $\mathsf{LPG}(s, b/\sqrt{d})$*, with $b > 0$ and $s \in \mathbb{N}^*$, then, conditionally on the fact that $m_0 \geq 5b/\sqrt{d}$, for any $0 < \varepsilon \leq \varepsilon_*$, we have*

$$\tau_{1/2}^+ \leq \begin{cases} d \cdot K/\varepsilon & \text{when } s = 1, \text{ and with the choice } \delta = \varepsilon/d \\ d \log^2(d) \cdot K/\varepsilon & \text{when } s = 2, \text{ and with the choice } \delta = \varepsilon/(d \log d) \\ d^{s-1} \cdot K/\varepsilon & \text{when } s \geq 3, \text{ and with the choice } \delta = \varepsilon d^{-s/2} \end{cases} \qquad (11)$$

*with probability at least $1 - K\varepsilon$, for generic constants $K, \varepsilon_* > 0$ that depend solely on the link function $\phi$ and the distribution $\nu$.*

Let us comment on this result. It says that that the integer $s$ coming from the growth condition controls the hardness of exiting the equator of the sphere. Indeed, as can be seen in $\mathsf{LPG}(s, b/\sqrt{d})$, the larger the $s$, the smaller the gradient projection is and hence the less information the SGD dynamics has to move from the initialization. This result can be seen as an extension of [Ben Arous et al., 2021, Theorem 1.3] valid only in the Gaussian case ($b = 0$). Furthermore, the Gaussian case shows that Theorem 3.2 is tight up to $\log(d)$ factors. Finally, note that the result is conditional to the fact that the

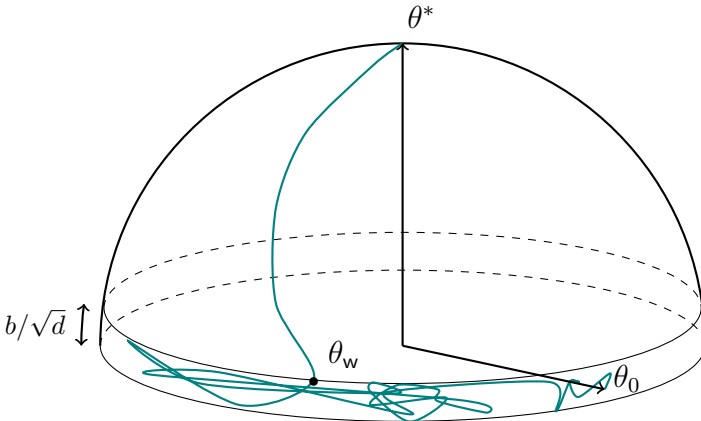

Figure 1: Sketch of the SGD dynamics. After a long time spent in a band of typical size $1/\sqrt{d}$, the dynamics escapes at *weak recovery* point $\theta_{\mathsf{w}}$ and then goes rapidly to the ground-truth $\theta^*$.

initialization is larger that some constant factor of $1/\sqrt{d}$, which has at least constant probability to happen in virtue of Lemma 2.1. This probability can be lowered by any constant factor by sampling offline a constant factor (independent of $d$) of i.i.d. initializations and keeping the one that maximizes its correlation with $\theta^*$. Finally, note that in all the cases covered by the analysis, it is possible to keep track of the constant $C$, and show that overall it depends only (i) on the property of $\nu$ w.r.t. the Gaussian on the one hand and (ii) on the Sobolev norm of the link function $\|\phi\|_{W_\nu^{1,4}}$ on the other.

**Strong recovery.** We place ourselves *after* the weak recovery time described in the previous question and want to understand if the $(\theta_t)_{t\geq 0}$ dynamics goes to $\theta^*$ and if yes, how fast it does so. This is what we refer to as *the strong recovery* question, captured by the fact that the one-dimensional summary statistics $m$ go towards 1. Thanks to the Markovian property of the SGD dynamics, we have the equality between all time $s > 0$ marginal laws of

$$\left(\theta_{\tau_{1/2}^+ + s} \,\Big|\, \tau_{1/2}^+,\, \theta_{\tau_{1/2}^+}\right) \overset{\text{Law}}{=} \left(\theta_s \,\Big|\, \theta_s = \theta_{\tau_{1/2}^+}\right),$$

and hence the strong recovery question is equivalent to study the dynamics with initialization that has already weakly recovered the signal, i.e. such that $m_\theta = 1/2$. We show that this part of the loss landscape is very different that the equator band in which the dynamics spends most of its times: in all the cases, we can choose stepsizes independent of the dimension and show that the time to reach the vicinity of $\theta^*$ will be independent of $d$.

**Theorem 3.3** (Strong Recovery). *Let* $(\theta_t)_{t\geq 0}$ *follow the SGD dynamics of Eq.(3) and let $L$ satisfy* $\mathsf{LPG}(s, b/\sqrt{d})$, *with $b > 0$ and $s \in \mathbb{N}^*$, then, for any $\varepsilon > 0$, taking $\delta = \varepsilon/d$, we have that there exists a time $T > 0$, such that*

$$|1 - m_T| \leq \varepsilon, \text{ and } \quad |T - \tau_{1/2}^+| \leq \mathsf{K} d \log(1/\varepsilon)\varepsilon^{-1} \tag{12}$$

*with probability at least $1 - \mathsf{K}\varepsilon$, for some generic $\mathsf{K} > 0$ that depends solely of the link function $\phi$.*

As introduced above, the important messages conveyed by this theorem are that (i) there is no difference between the different parameters setups captured by the information exponent $s$, and (ii) the time it takes to reach an $\varepsilon$-vicinity of $\theta^*$ is always strictly smaller than the one needed to exit the *weak recovery phase* (e.g. $d$ compared to $d^{s-1}$ when $s \geq 3$). This means that *the dynamics spends most of its time escaping the mediocrity*. Remark that we decided to present Theorem 3.3 resetting the step-size $\delta$ to put emphasis on the intrinsic difference between the two phases. Yet, we could have kept the same stepsize at the expense of slowing the second phase.

## 4 Typical cases of loss landscape with $\mathsf{LPG}$ property

In this section, we showcase two prototypical cases where the LPG holds true: the section 4.1 deals with the spherically symmetric setting, whereas the section 4.2 describes a perturbative regime where the distribution is *approximately* Gaussian is a quantitative sense.

## 4.1 The symmetric case

We start our analysis with the spherically symmetric setting. We show that a spherical harmonic decomposition provides a valid extension of the Hermite decomposition in the Gaussian case, leading to essentially the same quantitative performance up to constant (in dimension) factors.

**Spherical Harmonic Representation of the Population Loss.** We express the data distribution $\nu$ as a mixture of uniform measures $\nu = \int_0^\infty \tau_{r,d}\, \rho(dr)$, where $\tau_{r,d} = \mathrm{Unif}(r\mathcal{S}_{d-1})$. Let $\tau_d = \tau_{1,d}$ and $u_d \in \mathcal{P}([-1,1])$ be the projection of $\tau_d$ onto one direction, with density given in close form by $u_d(dt) = Z^{-1}(1-t^2)^{(d-3)/2}\mathbf{1}(|t| \leq 1)dt$, where $Z$ is a normarlizing factor. Let $\{P_{j,d}\}_{j\in\mathbb{N}}$ be the orthogonal basis of Gegenbauer polynomials of $L^2_{u_d}([-1,1])$, normalized such that $P_{j,d}(1) = 1$ for all $j,d$. For each $r > 0$, consider

$$l_r(\theta) := \langle \phi_\theta, \phi_{\theta^*}\rangle_{\tau_{r,d}} = \langle \phi_\theta^{(r)}, \phi_{\theta^*}^{(r)}\rangle_{\tau_d}\,,$$

where we define $\phi^{(r)} : [-1,1] \to \mathbb{R}$ such that $\phi^{(r)}(t) := \phi(rt)$. We write its decomposition in $L^2_{u_d}([-1,1])$ as $\phi^{(r)} = \sum_j \alpha_{j,r,d} P_{j,d}$, with $\alpha_{j,r,d} = \frac{\langle \phi^{(r)}, P_{j,d}\rangle}{\|P_{j,d}\|^2}$. Let $\Omega_d$ be the Lebesgue measure of $\mathcal{S}_{d-1}$, and $N(j,d) = \frac{2d+j-2}{d}\binom{d+j-3}{d-1}$ the so-called *dimension* on the spherical harmonics of degree $j$ in dimension $d$. From the Hecke-Funk representation formula [Frye and Efthimiou, 2012, Lemma 4.23] and the chosen normalization $P_{j,d}(1) = 1$, we have $\|P_{j,d}\|_2 = \left(\frac{\Omega_{d-1}}{\Omega_{d-2}N(j,d)}\right)^{1/2}$ [Frye and Efthimiou, 2012, Proposition 4.15] and obtain finally

$$l_r(\theta) = \sum_j \bar\alpha^2_{j,r,d} P_{j,d}(\theta \cdot \theta^*)\,, \tag{13}$$

where we defined for convenience $\bar\alpha_{j,r,d} = \alpha_{j,r,d}/\sqrt{N(j,d)}$. As a result, it follows that the overall loss writes as solely the correlaton $m_\theta = \theta \cdot \theta^*$ as

$$L(\theta) = 2\|\phi\|^2_{L^2_\nu} - 2\sum_j \beta_{j,d} P_{j,d}(m_\theta) := \ell(m)\,, \tag{14}$$

where $\beta_{j,d} = \int_0^\infty \bar\alpha^2_{j,r,d}\rho(dr) \geq 0$. Unsurprisingly, we observe that, thanks to the spherical symmetry and analogous to the Gaussian case, the loss still admits $m = \theta \cdot \theta^*$ as a summary statistics. Yet, it is represented in terms of Gegenbauer polynomials, rather than monomials as in the Gaussian case. The monomial representation is a consequence of the stability of the Gaussian measure, as seen by the fact that $\langle \phi_\theta, \phi_{\tilde\theta}\rangle_{\gamma_d} = \langle \phi, \mathsf{A}_{\theta \cdot \tilde\theta}\phi\rangle_\gamma$, where $(\mathsf{A}_m f)(t) = \mathbb{E}_{z\sim\gamma}[f(mt + \sqrt{1-m^2}z)]$ has a *semi-group* structure (it is even known in fact as the Ornstein-Ulhenbeck semi-group).

Let $\eta \in \mathcal{P}(\mathbb{R})$ be the marginal of $\nu$ along any direction. The following proposition charaterizes the coefficients $\beta_{j,d}$ as integrals over the radial distribution $\rho$ and projections of the link function $\phi$:

**Proposition 4.1** (Loss representation)**.** *The $\beta_{j,d}$ defined in* (14) *have the integral representation*

$$\beta_{j,d} = \langle \phi, \mathcal{K}_j\phi\rangle_{L^2_\eta}\,, \tag{15}$$

*where $\mathcal{K}_j$ is a positive semi-definite integral operator of $L^2_\eta$ that depends solely on $\rho$ and $\phi$.*

Note that a closed form expression of $\mathcal{K}_j$ can be found in Appendix D.2. The above proposition is in fact the stepping stone to calculate properly the *information exponent* that plays a crucial role in the property LPG. This is given through the link between the spectrum $\beta_{j,d}$ and the decomposition of $\phi$ in the $L^2_\eta$ orthogonal basis of polynomials, that we denote by $\{q_j\}_j$.

**Proposition 4.2.** *Let $s = \inf\{j; \beta_{j,d} > 0\}$ and $\tilde s = \inf\{j; \langle \phi, q_j\rangle_\eta \neq 0\}$. Then $s \leq \tilde s$.*

Thus, the number of vanishing moments of $\phi$ with respect to the data marginal $\eta$ provides an upper bound on the 'effective' information exponent of the problem $s$, as we will see next.

**Local Polynomial Growth.** From (14), and as $\nabla^{\mathcal{S}}_\theta L(\theta) = \ell'(m)(\theta^* - m\theta)$, we directly obtain

$$-\nabla^{\mathcal{S}}_\theta L(\theta) \cdot \theta^* = -(1-m^2)\ell'(m) = 2(1-m^2)\sum_j \beta_{j,d}P'_{j,d}(m)\,, \tag{16}$$

which is the quantity we want to understand to exhibit the property LPG in this case. Hence, we now turn into the question of obtaining sufficient guarantees on the coefficients $(\beta_{j,d})_j$ that ensure

local polynomial growth. Since the typical scale of initialization for $m$ is $\Theta(1/\sqrt{d})$, our goal is to characterize sufficient conditions of local polynomial growth with $b = O(1/\sqrt{d})$.

For that purpose, let us define two key quantities of Gegenbauer polynomials:

$$v_{j,d} := - \min_{t \in (0,1)} P_{j,d}(t) , \qquad \text{(smallest value)} \qquad (17)$$

$$z_{j,d} := \arg\max \{t \in (0,1); P_{j,d}(t) = 0\} . \qquad \text{(largest root)} \qquad (18)$$

We have the following sufficient condition based on the spectrum $(\beta_{j,d})_j$:

**Proposition 4.3** (Spectral characterization of LPG). *Suppose there exist constants $K, C > 0$ and $s \in \mathbb{N}$ such that we both have $\beta_{s,d} \geq C$ and $\sum_{j>s} \beta_{j,d} j(j+d-2) v_{j-1,d+2} \leq K d^{(3-s)/2}$ . Then, taking $s^*$ as the infimum of such $s$, $L$ has the property $\mathsf{LPG}(s^* - 1, z_{s^*,d})$. In particular, whenever $s^* \ll d$, we have $z_{s^*,d} \leq 2\sqrt{s^*/d}$.*

This proposition thus establishes that, modulo a mild regularity assumption expressed though the decay of the coefficients $\beta_{j,d}$, the spherically symmetric non-Gaussian setting has the same geometry as the Gaussian setting, for correlations *slightly* above the equator. Crucially, the required amount of correlation to 'feel' the local polynomial growth is a $O(\sqrt{s})$ factor from the typical initialization, and can be thus obtained with probability $\simeq e^{-s}$ over the initialization, according to Lemma 2.1, a lower bound which is *independent of $d$*.

The sufficient condition for $\beta_{j,d}$ appearing in Proposition 4.3 involves the minimum values $v_{j,d}$ of Gegenbauer polynomials $P_{j,d}$, as well as sums of the form $\sum_j j^2 \beta_{j,d}$. In order to obtain a more user-friendly condition, we now provide an explicit control of $v_{j,d}$, and leverage mild regularity of $\phi$ to control $\sum_j j^2 \beta_{j,d}$. This motivates the following assumption on $\phi$ and $\nu$:

**Assumption 4.4.** *The link function $\phi$ satisfies $\phi \in L^2_\eta$ and $\phi' \in L^4_\eta$, and the radial distribution $\rho$ has finite fourth moment $\mathbb{E}_\rho[r^4] < \infty$.*

**Theorem 4.5** (LPG for symmetric distributions). *Assume that $\phi$ and $\nu$ satisfy Assumption 4.4, and let $s^* = \inf\{j; \langle \phi, q_j \rangle_\eta \neq 0\}$. Then $L$ has the property $\mathsf{LPG}(s^* - 1, 2\sqrt{s^*/d})$.*

The proof is provided in Appendix D.5. At a technical level, the main challenge in proving Theorem 4.5 is to achieve a uniform control of $v_{j,d}$ in $j$, a result which may be of independent interest. We address it by combining state-of-the-art bounds on the roots of the Gegenbauer polynomials, allowing us to cover the regime where $j$ is small or comparable to $d$, together with integral representations via the Cauchy integral formula, providing control in the regime of large $j$. On the other hand, we relate the sum $\sum_j j^2 \beta_{j,d}$ to a norm of $\phi'$ using a Cauchy-Schwartz argument, where we leverage the fourth moments from Assumption 4.4.

**Remark 4.6.** *Since we are in a setting where $\phi$ is known, an alternative to the original recovery problem from Eq (2) is to consider a pure Gegenbauer 'student' link function of the form $\tilde{\phi} = P_{s,d}$, where $s$ is the information exponent from Proposition 4.2. Indeed, the resulting population loss $\tilde{L}(\theta) = \mathbb{E}[(\tilde{\phi}(x \cdot \theta) - \tilde{\phi}(x \cdot \theta^*))^2]$ satisfies the $\mathsf{LPG}$ property, as easily shown in Fact D.5.*

For the sake of completeness, we describe more precisely two concrete case studies below.

**Example 4.7** (Uniform Measure on the Sphere). *When $\nu = Unif(\sqrt{d}\mathcal{S}_{d-1})$, we have $\rho = \delta_{\sqrt{d}}$, and therefore $\beta_{j,d} = \bar{\alpha}^2_{j,\sqrt{d},d}$. In that case, the orthogonal polynomial basis $\{q_j(t)\}_j$ of $L^2_\eta$ coincides with the rescaled Gegenbauer polynomials, $q_j(t) = P_{j,d}(t/\sqrt{d})$. Consider now a link function $\phi$ with $s - 1$ vanishing moments with respect to $L^2_\eta$, i.e. such that $\bar{\alpha}_{j,d} = \langle \phi, q_j \rangle_\eta = 0$ for $j < s$ and $\bar{\alpha}_{s,d} = \langle \phi, q_s \rangle_\eta \neq 0$; and with sufficient decay in the higher harmonics as to satisfy the bound on the sum presented in Proposition 4.3 (for example, $\phi(t) = q_s(t)$ trivially satisfies this condition). Then Proposition 4.3 applies and we conclude that the resulting population landscape satisfies $\mathsf{LPG}(s - 1, O(\sqrt{s/d}))$.*

In Yehudai and Shamir [2020], Wu [2022] it is shown that monotonically increasing link functions[1] lead to a benign population landscape, provided the data distribution $\nu$ satisfies mild anti-concentration properties. We verify that in our framework.

---

[1]or link functions where their monotonic behavior dominates; see Wu [2022].

**Example 4.8** (Non-decreasing $\phi$). *Indeed, Proposition 4.3 is verified with $s = 1$, provided $\phi' \in L^2_\eta$. Indeed, if $\phi \neq 0$ is monotonic, then we have $\beta_1 = \langle \phi, \mathcal{K}_1 \phi \rangle_\eta = C_d \left( \mathbb{E}_\eta[t\phi(t)] \right)^2 \neq 0$, since we can assume without loss of generality that $\phi(t) \geq 0$ for $t \geq 0$ and $\phi(t) \leq 0$ for $t \leq 0$.*

We emphasize that the results of Yehudai and Shamir [2020], Wu [2022] extend beyond the spherically symmetric setting, which is precisely the focus of next section.

## 4.2  Non-Spherically Symmetric Case

We now turn to the setting where $\nu$ is no longer assumed to have spherical symmetry. By making further regularity assumptions on $\phi$, our main insight is that distributions that are *approximately* symmetric (defined in an appropriate sense) still benefit from a well-behaved optimization landscape.

**Two-dimensional Wasserstein Distance.** When $\nu$ is not spherically symmetric, the machinery of spherical harmonics does not apply, and we thus need to rely on another structural property. Consider a centered and isometric data distribution $\nu \in \mathcal{P}_2(\mathbb{R}^d)$, i.e. such that $\mathbb{E}_\nu x = 0$ and $\Sigma_\nu = \mathbb{E}_\nu[xx^\top] = I_d$. We consider the two-dimensional 1-Wasserstein distance [Niles-Weed and Rigollet, 2022, Definition 1] –see also Paty and Cuturi [2019]– between a pair of distributions $\nu_a, \nu_b \in \mathbb{P}(\mathbb{R}^d)$, defined as

$$\widetilde{W}_{1,2}(\nu_a, \nu_b) := \sup_{P \in \mathrm{Gr}(2,d)} W_1(P_\# \nu_a, P_\# \nu_b) \,, \tag{19}$$

where the supremum runs for any two-dimensional subspace $P \in \mathrm{Gr}(2, d)$, and $P_\# \nu \in \mathcal{P}(\mathbb{R}^2)$ is the projection (or marginal) of $\nu$ onto the span of $P$. $\widetilde{W}_{1,2}$ is a distance ([Paty and Cuturi, 2019, Proposition 1]) and measures the largest 1-Wasserstein distance over all two-dimensional marginals.

We are in particular interested in the setting where $\nu_a = \nu$ is our data distribution, and $\nu_b$ is a reference symmetric measure – for instance the standard Gaussian measure $\gamma_d$. Consider the fluctuations

$$\Delta_L(\theta) := |L(\theta) - \bar{\ell}(m_\theta)| \text{ and } , \tag{20}$$

$$\Delta_{\nabla L}(\theta) := |\nabla^{\mathcal{S}}_\theta L(\theta) \cdot \theta^* - \bar{\ell}'(m_\theta)(1 - m_\theta^2)| \,, \tag{21}$$

where $\bar{\ell}(m)$ is the Gaussian loss defined in Section 2 and $L(\theta) = \mathbb{E}_\nu[|\phi(x \cdot \theta) - \phi(x \cdot \theta^*)|^2]$. $\Delta_L$ and $\Delta_{\nabla L}$ thus measure respectively the fluctuations of the population loss and the relevant (spherical) gradient direction. By making additional mild regularity assumptions on the link function $\phi$, we can obtain a uniform control of the population loss geometry using the dual representation of the 1-Wasserstein distance.

**Assumption 4.9** (Regularity of link function). *We assume that $\phi, \phi'$ are both $B$-Lipschitz, and that $\phi''(t) = O(1/t)$.*

**Assumption 4.10** (Subgaussianity). *The data distribution $\nu$ is $M$-subgaussian: for any $v \in \mathcal{S}_{d-1}$, we have $\|x \cdot v\|_{\psi_2} \leq M$, where $\|z\|_{\psi_2} := \inf\{t > 0; \mathbb{E}[\exp(z^2/t^2) \leq 2\}$ is the Orlitz-2 norm.*

**Proposition 4.11** (Uniform gradient approximation). *Under Assumptions 4.9 and 4.10, for all $\theta \in \mathcal{S}_{d-1}$,*

$$\Delta_{\nabla L}(\theta) = (1 - m^2) O\left( \widetilde{W}_{1,2}(\nu, \gamma) \log(\widetilde{W}_{1,2}(\nu, \gamma)^{-1}) \right) \tag{22}$$

*where the $O(\cdot)$ notation only hides constants appearing in Assumptions 4.9 and 4.10.*

In words, the population gradient under $\nu$ is viewed as a perturbation of the population gradient under $\gamma$, which has the well-behaved geometry already described in Section 2. These perturbations can be uniformly controlled by the projected 1-Wasserstein distance, thanks to the subgaussian tails of $\nu$.

Our focus will be in situations where $\widetilde{W}_{1,2}(\nu, \gamma) = O(1/\sqrt{d})$. This happens to be the 'natural' optimistic scale for this metric in the class of isotropic distributions $\Sigma_\nu = I_d$, as can be seen for instance when $\nu = \mathrm{Unif}(\sqrt{d}\mathcal{S}_{d-1})$. Under such conditions, it turns out that link functions with information exponent $s \leq 2$ can be recovered with simple gradient-based methods, by paying an additional polynomial (in $d$) cost in time complexity.

**Assumption 4.12.** *The Gaussian information exponent of $\phi$, $s := \arg\min\{j; \langle \phi, H_j \rangle \neq 0\}$ satisfies $s \leq 2$.*

**Assumption 4.13.** *The projected Wasserstein distance satisfies* $\widetilde{W}_{1,2}(\nu,\gamma) \leq M'/\sqrt{d}$.

**Proposition 4.14** (LPG, non-symmetric setting). *Under Assumptions 4.9, 4.10, 4.12 and 4.13, L verifies* $\mathsf{LPG}\left(1, O\left(\sqrt{\frac{(\log d^\kappa)}{d}}\right)\right)$, *where $\kappa$ depends only on $B, M, M'$.*

This proposition illustrates the cost of breaking spherical symmetry in two aspects: (i) it requires additional regularity on $\phi$, and notably restricts its (Gaussian) information exponent to $s = 2$, and (ii) the scale to reach LPG is now no longer dimension-free, but has a polynomial dependency on dimension, since from Lemma 2.1, picking $\delta$ any positive constant we have

$$\mathbb{P}_{\theta_0}\left(m_{\theta_0} \geq \sqrt{\log d^\kappa}/\sqrt{d}\right) \geq \frac{\delta}{4}e^{-(\sqrt{\log d^\kappa}+\delta)^2} \geq \Omega\left(d^{-(1+o(1))\kappa}\right) .$$

We are not able to rule this out as a limitation of our proof; whether this polynomial dependency on dimension is inherent to the symmetry breaking is an interesting question for future work.

While assumptions 4.9, 4.10 and 4.12 are transparent and impose only mild conditions on the link function and tails of $\nu$, the 'real' assumption of Proposition 4.14 is the concentration of $\widetilde{W}_{1,2}(\nu,\gamma)$ (Assumption 4.13). The ball $\{\nu; \widetilde{W}_{1,2}(\nu,\gamma) = O(1/\sqrt{d})\}$ contains many non-symmetric measures, for instance empirical measures sampled from $\gamma$ with $n = \omega(d^2)$ [Niles-Weed and Rigollet, 2022, Proposition 8], and we suspect it contains many other examples, such as convolutions of the form $\nu * \gamma_\sigma$ arising for instance in diffusion models. That said, one should *not* expect the distance $\widetilde{W}_{1,2}(\nu,\gamma)$ to be of order $1/\sqrt{d}$ for generic 'nice' distributions $\nu$; for instance, log-concave distributions are expected to satisfy $W_1(P_\#\nu,\gamma) \simeq 1/\sqrt{d}$ for *most* subspaces $P$, as captured in Klartag's CLT for convex bodies Klartag [2007].

**Localized Growth via Stein's method.** To illustrate the mileage of the previous techniques beyond this 'quasi-symmetric' case, we consider now an idealised setting where the data is drawn from a product measure $\nu = \eta^{\otimes d}$, with $\eta \in \mathcal{P}(\mathbb{R})$ and $W_1(\eta,\gamma_1) = \Theta(1)$. In other words, $x = (x_1,\ldots,x_d) \sim \nu$ if $x_i \sim \eta$ are i.i.d. In this setting, the distances $W_1(P_\#\nu,\gamma)$ reflect a CLT phenomena, which requires the subspace $P$ to 'mix' across independent variables. Consequently, one may expect the expression of the hidden direction $\theta^*$ in the canonical basis to play a certain role so we make the following additional regularity assumption on the tails of $\phi$:

**Assumption 4.15** (Additional Regularity in third derivatives). *$\phi$ admits four derivatives bounded by $L$, with $|\phi^{(3)}(t)| = O(1/t)$ and $|\phi^{(4)}(t)| = O(1/t^2)$. Moreover, the third moment of the data distribution is finite: $\tau_3 = \mathbb{E}_{t\sim\eta}[t^3] < \infty$.*

Stein's method provides a powerful control on $\Delta_L(\theta)$ and $\Delta_{\nabla L}(\theta)$, as shown by the following result:

**Proposition 4.16** (Stein's method for product measure). *Let $\chi(\theta,\theta^*) := \|\theta\|_4^2 + \|\theta^*\|_4^2$. Under Assumptions 4.9, 4.10 and 4.15, there exists a universal constant $C = C(M, B, \tau_3)$ such that*

$$\Delta_L(\theta) \leq C\chi(\theta,\theta^*) \,, and \ \Delta_{\nabla L}(\theta) \leq C\sqrt{1-m^2}\chi(\theta,\theta^*) \,. \tag{23}$$

The proof is based on the Stein's method for multivariate variables [Röllin, 2013, Theorem 3.1] with independent entries, which provides a quantitative CLT bound. Contrary to the quasi-symmetric case, here the concentration is not uniform over the sphere, but crucially depends on the sparsity of *both* $\theta$ and $\theta^*$, measured via the $\ell_4$ norms $\|\theta\|_4, \|\theta^*\|_4$: for incoherent, non-sparse directions, we have $\|\theta\|_4^2 \simeq 1/\sqrt{d}$, recovering the concentration rate that led to Proposition 4.14, while for sparse directions we have $\|\theta\|_4^2 = \Theta(1)$, indicating an absence of concentration to the Gaussian landscape.

Therefore, the natural conclusion is to assume a planted model where $\theta^*$ is *incoherent* with the data distribution, i.e. $\|\theta^*\|_4 = O(d^{1/4})$. While the LPG property does not directly apply in this setting, we outline an argument that suggests that the single-index model can still be efficiently solved using gradient-based methods. For that purpose, we assume that $\theta^*$ is drawn uniformly in $\mathcal{S}_{d-1}$, which implies that its squared-$L^4$ norm $\|\theta^*\|_4^2$ is of order $d^{-1/2}$ with high probability:

**Fact 4.17.** *Assume $\theta^* \sim \text{Unif}(\mathcal{S}_{d-1})$. Then $\mathbb{P}(\|\theta^*\|_4^2 \leq C/\sqrt{d}) \geq 1 - C' \exp(-C)$.*

Because $\theta_0$ is also drawn uniformly on the sphere, the typical value of $\chi(\theta,\theta^*)$ is of order $d^{-1/2}$. For Gaussian information exponent $s = 2$, the population gradient under Gaussian data satisfies

$-\nabla_\theta^{\mathcal{S}_{d-1}} L_\gamma(\theta) \cdot \theta^* \geq C m_\theta$. As a consequence, by Proposition 4.16, whenever $|\theta_0 \cdot \theta^*| > c\sqrt{d}$ (which happens with constant probability lower bounded by $e^{-c}$), we enter a 'local' LPG region where $-\nabla_\theta^{\mathcal{S}_{d-1}} L(\theta) \cdot \theta^* \geq C(m - c/\sqrt{d}) > 0$. While this condition is sufficient in the quasi-symmetric setting to start accumulating correlation (Theorem 3.2), now this event is conditional on $\theta$ being dense, ie so that $\chi(\theta, \theta^*) = O(1/\sqrt{d})$.

Since the typical value of $\chi(\theta, \theta^*)$ is of scale $1/\sqrt{d}$, one would expect that SGD will rarely visit sparse points where $\chi(\theta, \theta^*) \gg O(1/\sqrt{d})$, and thus that the local LPG property will be valid for *most* times during the entropic phase of weak recovery, summarized by the following conjecture:

**Conjecture 4.18** (SGD avoids sparse points). *Assume $\theta^*, \theta_0$ are drawn from the uniform measure, and let $\theta_t$ be the $t$-th iterate of SGD with $\delta \simeq 1/(d \log d)$. There exists a universal constant $C$ such that for any $\xi > 0$, we have*

$$\mathbb{P}\left(\sup_{t \leq T} \|\theta_t\|_4^2 \geq \sqrt{\frac{\xi \log T}{d}}\right) \leq C \exp(-\xi^2 d) . \tag{24}$$

Since the time to escape mediocrity in the case $s = 2$ is $T \simeq d \log(d)^2$, this conjecture would imply that SGD does not effectively 'see' any sparse points, and thus escapes mediocrity. If one assumed that in this phase the dynamics is purely noisy, now pretending that $\theta_i$ were drawn independently from the uniform measure, and that $\|\theta\|_4^4$ is approximately Gaussian with mean $d^{-1}$ and variance $d^{-3}$, the result follows by simple concentration. The challenging aspect of Conjecture 4.18 is precisely to handle the dependencies across iterates, as well as the spherical projection steps.

# 5 Experiments

In order to validate our theory, and inspect the degree to which our bounds may be pessimistic, we consider empirical evaluation of the training process in our two primary settings. Specifically, we consider random initialization on the half-sphere (with the sign chosen to induce positive correlation as in Arous et al. [2021]), and investigate how often strong recovery occurs relative to the information exponent of the link function. Ultimately the empirical evidence corroborates the theory, showing the symmetric case succeeds when the initial correlation is above a polynomial threshold and the non-symmetric case succeeds as long as $s = 2$. Due to space constraints, the details of the experimental setup and further discussion appear in Section A.

# 6 Conclusion and Perspectives

In this work, we have asked whether the remarkable properties of high-dimensional Gaussian SGD regression of single-index models are preserved as one loses some key aspects that make Gaussian distributions so special (and so appealing for theorists). Our results are mostly positive, indicating a robustness of the Gaussian theory, especially within the class of spherically symmetric distributions, where a rich spherical harmonic structure is still available. As one loses spherical symmetry, the situation becomes more dire, motivating a perturbative analysis that we have shown is effective via projected Wasserstein and Stein couplings.

That said, there are several open and relevant avenues that our work has barely touched upon, such as understanding whether the robustness can be transferred to other algorithms beyond SGD, or addressing the semi-parametric problem when the link function is unknown, along the lines of Bietti et al. [2022], Abbe et al. [2023], Damian et al. [2022], Berthier et al. [2023]. A particularly interesting direction of future work is to extend the analysis of product measures to 'weakly dependent' distributions, motivated by natural images where locality in pixels captures most (but not all) of the statistical dependencies. Stein's method appears to be a powerful framework that can accommodate such weak dependencies, and deserves future investigation.

**Acknowledgements** This work was partially supported by NSF DMS 2134216, NSF CAREER CIF 1845360, NSF IIS 1901091 and the Alfred P Sloan Foundation.

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
