# Appendices

We gather in the appendix the proofs of the theorems, propositions and lemmas stated in the main text. In Section B, the reader will find a short proof of Lemma 2.1. In Section C, we prove Theorems 3.2 and 3.3 on the SGD dynamics. Sections D and E are reciprocally devoted to prove that the LPG property holds in some spherical symmetric case and under some perturbative regime. In Section A we include the experimental figures from the experiments run in the main text.

## A  Experimental Figures and Discussion

**Symmetric Case**    For the spherically symmetric setting, we experiment with the input distribution that is uniform on the sphere. We are interested in verifying that, unlike the Gaussian case, strong recovery depends on whether the initial correlation is sufficiently high to avoid local minima and benefit from the LPG guarantee. This is not evident in the 2nd degree Gegenbauer case because it is monotonic, but is clear from the 4th degree Gegenbauer link function.

In the infinite sample setting, Figure 3 exactly characterizes the loss landscape when learning the 4th degree Gegenbauer under inputs uniform on $\mathcal{S}_{d-1}$ for different values of $d$. Note that the largest zero for $d = 50$ occurs at $\approx \pm 0.31$, and the loss is monotonic for $m$ values initialized outside that region. This phenomenon persists for higher dimensions, and one may observe that $d$ increases, the critical points become smaller in magnitude, according to the scaling $\simeq \sqrt{1/d}$.

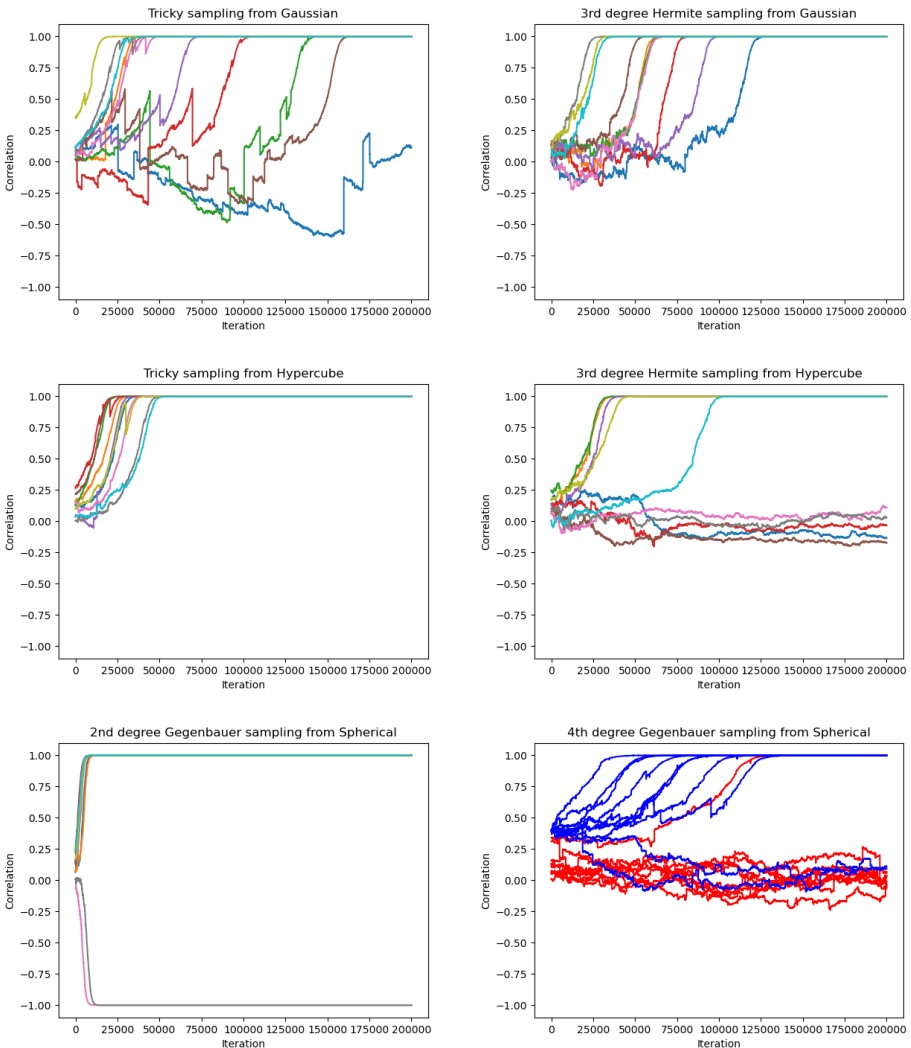

Figure 2: Correlation with the true signal throughout training, under different choices of link function and input distribution.

The bottom right subplot in Figure 2 indicates training runs in this setting, where red lines are initialized uniformly on the sphere, and blue lines are initialized uniformly conditioned on $m = 0.4$, which is slightly past the last zero of the polynomial. We observe that random initialization infrequently exceeds the threshold necessary for strong recovery, but planting the initialization above this threshold gives a high probability of recovery.

**Non-Symmetric Case** For the non-spherically symmetric setting, we compare the performance of Gaussian inputs with inputs that are approximately Gaussian under a two-dimensional projection. For simplicity, we loosen our assumptions slightly, and consider the input distribution as the $d$ dimensional product distribution of uniform random variables (rescaled to have unit variance), and allow for a non-Lipschitz link function. Here, we are primarily interested in whether Assumption 4.12 is tight and $s \leq 2$ is necessary for recovery, as well as whether Conjecture 4.18 holds in practice.

To evaluate, we compare strong recovery rates when training on a "tricky" function with $s = 2$ (chosen to be $\frac{1}{2}(h_2 - h_3 - h_4 + h_5)$) versus a function with $s = 3$ (simply the degree three hermite polynomial $h_3$). We make this choice for the $s = 2$ function in order to produce a function which is not monotonic, for which learning is guaranteed as discussed in Yehudai and Shamir [2020].

In Figure 2 we observe that strong recovery reliably occurs for both the Gaussian and hypercube input distributions when $s = 2$. There is more variance in the Gaussian runs, likely because the magnitude

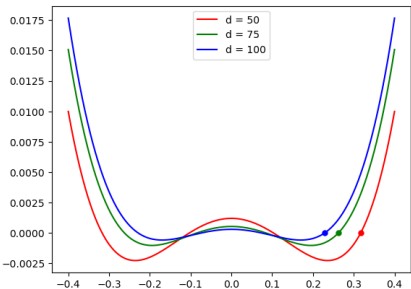

Figure 3: The 4th degree Gegenbauer polynomial with dimension 50, 75, 100. Each is equivalent (up to rescaling) to the loss landscape of learning the 4th degree Gegenbauer in the appropriate dimension. Points indicate largest zeros.

of the gradients will be larger due to the inclusion of high degree terms. But for $s = 3$, the Gaussian distribution converges quickly while the hypercube distribution often cannot escape the equator.

# B    Proof of Lemma 2.1

Let us first recall the Lemma before writing a proof of it.

**Lemma.** *For all $a > 0$, we have $\mathbb{P}_{\theta_0}(m_{\theta_0} \geq a/\sqrt{d}) \leq a^{-1}e^{-a^2/4}$. Additionally, for any $\delta > 0$ such that $\max\{a, \delta\} \leq \sqrt{d}/4$, we have the lower bound: $\mathbb{P}_{\theta_0}(m_{\theta_0} \geq a/\sqrt{d}) \geq \frac{\delta}{4}e^{-(a+\delta)^2}$.*

*Proof.* By rotation invariance of the uniform distribution of the sphere, $m_{\theta_0}$ is distributed according to $\theta_0[1]$, the first coordinate of the vector $\theta_0 \in \mathcal{S}_{d-1}$. By a particular case of Stam's formula [Stam, 1982, relation (3)], we know that for $d \geq 3$, both are distributed according to the probability of density, $\forall t \in \mathbb{R}$,

$$\tau(t) := \frac{\Gamma(d/2)}{\sqrt{\pi}\Gamma((d-1)/2)} \left(1 - t^2\right)^{(d-3)/2} \mathbb{1}_{[-1,1]}.$$

First, note that we can upper and lower bound the constant by the following:

$$\sqrt{\frac{d}{3}} \leq \frac{\Gamma(d/2)}{\Gamma((d-1)/2)} \leq \sqrt{\frac{d}{2}},$$

for $d \geq 6$ by [Laforgia and Natalini, 2013, equality 3.2], which was already proved in Gautschi [1959].

Hence, in terms of the upper bound, we have:

$$\begin{aligned}
\mathbb{P}_{\theta_0}(m_{\theta_0} \geq a/\sqrt{d}) &\leq \sqrt{\frac{d}{2\pi}} \int_{a/\sqrt{d}}^{1} \left(1 - t^2\right)^{(d-3)/2} dt \\
&\leq \sqrt{\frac{d}{2\pi}} \int_{a/\sqrt{d}}^{1} e^{-\frac{d-3}{2}t^2} dt \\
&\leq \frac{1}{\sqrt{2\pi}} \frac{d}{a} \int_{a/\sqrt{d}}^{1} t e^{-\frac{d-3}{2}t^2} dt \\
&\leq \frac{1}{2a}e^{-a^2/4},
\end{aligned}$$

which concludes the first part of the result.

Second, let $a \leq \sqrt{d}/4$ and take any $0 < \delta < \sqrt{d}/4$. We have,

$$\mathbb{P}_{\theta_0}(m_{\theta_0} \geq a/\sqrt{d}) \geq \sqrt{\frac{d}{3\pi}} \int_{a/\sqrt{d}}^{1} \left(1 - t^2\right)^{(d-3)/2} dt$$

$$\geq \sqrt{\frac{d}{3\pi}} \int_{a/\sqrt{d}}^{(a+\delta)/\sqrt{d}} \left(1 - t^2\right)^{(d-3)/2} dt$$

$$\geq \sqrt{\frac{d}{3\pi}} \frac{\delta}{\sqrt{d}} \left(1 - \frac{(a+\delta)^2}{d}\right)^{(d-3)/2},$$

where the last inequality simply comes from the fact that $t \to (1-t^2)^{(d-3)/2}$ is non-increasing. Going further, if we lower bound the term with the negative $-3/2$ power by 1, we have

$$\mathbb{P}_{\theta_0}(m_{\theta_0} \geq a/\sqrt{d}) \geq \frac{\delta}{4} \exp\left(\frac{d}{2} \log\left(1 - \frac{(a+\delta)^2}{d}\right)\right)$$

$$\geq \frac{\delta}{4} \exp\left(-\frac{(a+\delta)^2}{2\left(1 - (a+\delta)^2/d\right)}\right),$$

where the last inequality come from the classical bound $\log(1+x) \geq x/(1+x)$, for $x > -1$. Furthermore, as, $(a+\delta)^2 \leq d/2$, we have finally

$$\mathbb{P}_{\theta_0}(m_{\theta_0} \geq a/\sqrt{d}) \geq \frac{\delta}{4} e^{-(a+\delta)^2},$$

which finalizes the proof of the Lemma. ∎

## C  Proofs on the SGD dynamics: Section 3

We first recall the notations useful to fully describe the dynamics. In Section C.3, we prove Theorem 3.2 about weak recovery. In Section C.4, we prove Theorem 3.3 about strong recovery. Finally,

### C.1  Recalling the dynamics

For the sake of clarity, let us recall the notations and facts developed in the main text. The overall loss classically corresponds to the average over all the *data* of a square penalisation $l(\theta, x) = (\phi_\theta(x) - \phi_{\theta^*}(x))^2$ so that

$$L(\theta) = \mathbb{E}_\nu[(\phi_\theta(x) - \phi_{\theta^*}(x))^2].$$

To recover the signal given by $\theta^*$, we run *online stochastic gradient descent* on the sphere $\mathcal{S}_{d-1}$. This corresponds to have at each iteration $t \in \mathbb{N}^*$ a *fresh sample* $x_t$ independent of the filtration $\mathcal{F}_t = \sigma(x_1, \ldots, x_{t-1})$ and perform a spherical gradient step, with step-size $\delta > 0$, with respect to $\theta \to l(\theta, x_t)$:

$$\theta_{t+1} = \frac{\theta_t - \delta \nabla_\theta^{\mathcal{S}} l(\theta_t, x_t)}{\left|\theta_t - \delta \nabla_\theta^{\mathcal{S}} l(\theta_t, x_t)\right|}, \tag{25}$$

initialized at $\theta_0$ uniformly on the sphere: $\theta_0 \sim \text{Unif}(\mathcal{S}_{d-1})$. Recall that we use the notation $\nabla_\theta^{\mathcal{S}}$ to denote the spherical gradient, that is

$$\nabla_\theta^{\mathcal{S}} l(\theta, x) = \nabla_\theta l(\theta, x) - (\nabla_\theta l(\theta, x) \cdot \theta)\theta.$$

Let us introduce the following frequently used notations: for all $t \in \mathbb{N}^*$, we denote the normalization by $r_t := r(\theta_t, x_t) = \left|\theta_t - \delta \nabla_\theta^{\mathcal{S}} l(\theta_t, x_t)\right|$ and the martingale induced by the stochastic gradient descent as $M_t = M(\theta_t, x_t) = l(\theta_t, x_t) - \mathbb{E}_\nu[l(\theta_t, x)]$.

### C.2  Tracking the correlation.

Recall that the relevant signature of the dynamics is the one-dimensional correlation: $m_t = \theta_t \cdot \theta^*$. Let us re-write the iterative recursion followed by $(m_t)_{t \geq 0}$, with the notation recalled above, for $t \in \mathbb{N}^*$,

$$m_{t+1} = \frac{1}{r_t}\left(m_t - \delta \nabla^S l(\theta_t, x_t) \cdot \theta^*\right) = \frac{1}{r_t}\left(m_t - \delta \nabla^S L(\theta_t) \cdot \theta^* - \delta \nabla^S M_t \cdot \theta^*\right). \tag{26}$$

We want to lower bound the right hand side of (26). We begin by a lower bound on $1/r_t$.

**Lemma C.1** (Bound on $r_t$). *For all $t \in \mathbb{N}^*$, we have $1/r_t \geq 1 - \delta^2 \left| \nabla_\theta l(\theta_t, x_t) \right|^2$.*

*Proof.* For all $t \in \mathbb{N}^*$, we have, by orthogonality of as $\theta_t$ and $\nabla_\theta^{\mathcal{S}} l(\theta_t, x_t)$, that

$$r_t^2 = \left| \theta_t - \delta \nabla_\theta^{\mathcal{S}} l(\theta_t, x_t) \right|^2 = 1 + \delta^2 \left| \nabla_\theta^{\mathcal{S}} l(\theta_t, x_t) \right|^2 \leq 1 + \delta^2 \left| \nabla_\theta l(\theta_t, x_t) \right|^2.$$

Hence, from the inequality $(1 + u)^{-1/2} \geq 1 - u$ for all $u > 0$, we conclude the proof. ∎

Thanks the fact that $L$ satisfies $\mathsf{LPG}(s, b/\sqrt{d})$, ie $-\nabla^{\mathcal{S}} L(\theta) \cdot \theta^* \geq C(1 - m)(m - b/\sqrt{d})^{s-1}$, we have that the dynamics satisfies the following inequality between iterates:

$$m_{t+1} \geq m_t + C\delta(1 - m_t)\left(m_t - \frac{b}{\sqrt{d}}\right)^{s-1} - \delta \nabla^S M_t \cdot \theta^* - \delta^2 |m_t| \left| \nabla_\theta l(\theta_t, x_t) \right|^2 - \delta^3 \xi_t, \tag{27}$$

where $\xi_t = \left| \nabla_\theta l(\theta_t, x_t) \right|^2 \left| \nabla^S l(\theta_t, x_t) \cdot \theta^* \right|$. All the terms of the inequality have a natural origin: the second term is the ideal term coming from the gradient flow and the growth condition, the third term corresponds to the martingale increments coming form the noise induced by SGD and the two final terms are simply discretization errors coming from discrete nature of the procedure and the projection step.

However, to have a tight dependency with respect to the dimension, we need to be extra careful. This is why, following Arous et al. [2021], we decompose this term introducing a threshold $\mathsf{M} > 0$, to be fixed later, such that:

$$|m_t| \left| \nabla_\theta l(\theta_t, x_t) \right|^2 = |m_t| \left| \nabla_\theta l(\theta_t, x_t) \right|^2 \mathbb{1}_{\{|\nabla_\theta l(\theta_t, x_t)|^2 \leq \mathsf{M}\}} + |m_t| \left| \nabla_\theta l(\theta_t, x_t) \right|^2 \mathbb{1}_{\{|\nabla_\theta l(\theta_t, x_t)|^2 > \mathsf{M}\}}$$

With the same notations and summing all these terms until time $T \in \mathbb{N}^*$, we can write

$$m_T \geq m_0 + C\delta \sum_{t=0}^{T-1}(1 - m_t)(m_t - b/\sqrt{d})^{s-1} - \delta \sum_{t=0}^{T-1} \nabla^S M_t \cdot \theta^* - \delta^2 \sum_{t=0}^{T-1} |m_t| \left| \nabla_\theta l_t \right|^2 \mathbb{1}_{\{|\nabla_\theta l_t|^2 \leq \mathsf{M}\}}$$

$$- \delta^2 \sum_{t=0}^{T-1} |m_t| \left| \nabla_\theta l_t \right|^2 \mathbb{1}_{\{|\nabla_\theta l_t|^2 > \mathsf{M}\}} - \delta^3 \sum_{t=0}^{T-1} \xi_t,$$

where we use, for the sake of compactness, the shortcut notation $l_t = l(x_t, \theta_t)$. The strategy of the proof is the following: the first term is the drift term that makes the correlation grow, the second term is simply a martingale term that we deal with via standard martingale inequality, and the forth and fifth term are discretization error that we will bound loosely. The difficulty comes from the third term: the proof is based on the fact that we use a "part" of the drift term (say half) to control it. This is why we decide to rewrite finally our inequality as,

$$m_T \geq m_0 + \delta \frac{C}{2} \sum_{t=0}^{T-1}(1 - m_t)(m_t - b/\sqrt{d})^{s-1} - \delta \sum_{t=0}^{T-1} \nabla^S M_t \cdot \theta^* - \delta \sum_{t=0}^{T-1} D_t \tag{28}$$

$$- \delta^2 \sum_{t=0}^{T-1} |m_t| \left| \nabla_\theta l_t \right|^2 \mathbb{1}_{\{|\nabla_\theta l_t|^2 > \mathsf{M}\}} - \delta^3 \sum_{t=0}^{T-1} \xi_t,$$

where we have defined $D_t := \frac{C}{2}(1 - m_t)(m_t - b/\sqrt{d})^{s-1} - \delta |m_t| \left| \nabla_\theta l_t \right|^2 \mathbb{1}_{\{|\nabla_\theta l_t|^2 \leq \mathsf{M}\}}$. The following section show how to control these five terms in a quantitative way.

### C.3 Weak recovery

**Good initialization.** *During all this section, we condition on the event $\{m_0 \geq 5b/\sqrt{d}\}$.*

Before stating these lemmas, let us introduce some new notations. As already introduce, we recall that we denote $S_\eta := \{\theta \in \mathcal{S}_{d-1}, m_\theta \geq \eta\}$, the spherical cap of level $\eta \in (0, 1)$. Moreover for $\alpha \in (-1, 1)$, similarly to what is done in Ben Arous et al. [2021], we define the following stopping times $\tau_\alpha^+ := \inf\{t \geq 0, m_{\theta_t} \geq \alpha\}$ and $\tau_\alpha^- := \inf\{t \geq 0, m_{\theta_t} \leq \alpha\}$ reciprocally as the first time when $(\theta_t)_{t \geq 0}$ enters in $S_\alpha$ or leaves $S_\alpha$.

### C.3.1 Proof of Theorem 3.2

Thanks to Lemmas C.2, C.3, C.4, C.5 and C.6, that serve bounding all the terms in the $m_T$ inequality, there exists a constant $K$ that depend solely on the model such that we have the following lower bound: for all $\lambda > 0$, conditionally to the event on the events $\{T \leq \tau_{1/2}^+ \wedge \tau_{2b/\sqrt{d}}^-\}$,

$$m_T \geq m_0 + \frac{C}{2^{s+1}}\delta \sum_{t=0}^{T-1} m_t^{s-1} - 4\lambda,$$

with probability larger that $1 - \left(\frac{KT\delta^2}{\lambda^2} + \exp\left(-\frac{\lambda^2}{2K^2\delta^2 T + \lambda\delta(C + \delta\mathsf{M})}\right) + \frac{KTd^2\delta^2}{\lambda\mathsf{M}} + \frac{KTd\delta^3}{\lambda}\right).$

Now we choose $\lambda = b/\sqrt{d}$ and $\mathsf{M} = d^{3/2}$ so that

$$m_T \geq \frac{b}{\sqrt{d}} + \frac{C}{2^{s+1}}\delta \sum_{t=0}^{T-1} m_t^{s-1},$$

with probability at least $1 - p_{\delta,M}(T)$, where we defined naturally

$$p_{\delta,M}(T) := \left(\frac{KTd\delta^2}{b^2} + \exp\left(-\frac{b^2}{2K^2d\delta^2 T + b\sqrt{d}\delta(C + \delta\mathsf{M})}\right) + \frac{KTd^{5/2}\delta^2}{b\mathsf{M}} + \frac{KTd^{3/2}\delta^3}{b}\right).$$

Let us upper bound the probability $p_{\delta,M}(T)$. Let us set $\varepsilon > 0$ a small constant. First, in the exponential term, the term $b\sqrt{d}\delta(C + \delta d^{3/2})$ is negligible in virtue of the fact that in any of the cases of Theorem 3.2, we have $\delta \leq \varepsilon/d$. Moreover, for the sake of clarity, we gather all constant $K, C, b$ as one constant generic $\mathsf{K}$, as these depend only on the data distribution and the link function. Hence, for $d$ large enough,

$$p_{\delta,M}(T) \leq \mathsf{K}\left(dT\delta^2 + \exp\left(-\frac{1}{dT\delta^2}\right) + dT\delta^2 + d^{3/2}T\delta^3\right),$$

and as $d^{3/2}T\delta^3 \lesssim dT\delta^2$ for the range of $\delta$ we choose, we have $p_{\delta,M}(T) \leq \mathsf{K}\left(dT\delta^2 + \exp\left(-\frac{1}{dT\delta^2}\right)\right)$, and considering that we will take in any case $dT\delta^2 \leq 1$, as we have the inequality $\exp\left(-\frac{1}{dT\delta^2}\right) \leq dT\delta^2$, so that finally

$$p_{\delta,M}(T) \leq \mathsf{K}dT\delta^2$$

We divide the proof into the three cases $s = 1$, $s = 2$, $s \geq 3$.

**Case $s = 1$, $\delta = \varepsilon/d$.** In this case, we have that with probability $1 - p_{\delta,M}(T)$,

$$m_T \geq \frac{b}{\sqrt{d}} + \frac{C\delta}{2^s}T.$$

The right and side is larger than $1/2$ as soon as $\delta T \geq 2^s/C$. From this we have that with probability at least $1 - p_{\delta,M}(T)$, the hitting time is upper bounded by

$$\tau_{1/2}^+ \leq \frac{2^s}{C\delta}.$$

Now, taking $\delta = \varepsilon d^{-1}$, we can check that for $\varepsilon$ small enough, $dT\delta^2 \leq 2^s\varepsilon/C = \varepsilon\mathcal{O}(1)$ so that we have that with probability at least $1 - \mathsf{K}\varepsilon$, we have

$$\tau_{1/2}^+ \leq \frac{\mathsf{K}}{\varepsilon}d.$$

**Case $s = 2$, $\delta = \varepsilon/(d\log d)$.** Now by a discrete version of Grönwall inequality, recalled in Lemma C.10, we have with probability at least $1 - p_{\delta,M}(T)$,

$$m_T - \frac{b}{\sqrt{d}} \geq \frac{b}{\sqrt{d}}\left(1 + \delta\frac{C}{2}\right)^T \geq \frac{b}{\sqrt{d}}e^{C\delta T},$$

for $d$ large enough. And as the right hand side is larger than $1/2 + b/\sqrt{d}$ whenever,

$$\delta T \geq \frac{1}{C} \log(\sqrt{d}/4b),$$

for $d$ large enough compared to $b$. Then taking such a $T$, with probability at least $1 - p_{\delta,M}(T)$, the hitting time is upper bounded by

$$\tau_{1/2}^+ \leq \frac{2}{C\delta} \log(d).$$

Now, taking $\delta = \varepsilon d^{-1}(\log d)^{-1}$, we can check that for $\varepsilon$ small enough, $dT\delta^2 \leq \frac{\varepsilon}{C} = \varepsilon\mathcal{O}(1)$ so that we have that with probability at least $1 - \mathsf{K}\varepsilon$, we have

$$\tau_{1/2}^+ \leq \frac{\mathsf{K}}{\varepsilon} d \log(d)^2.$$

**Case** $s \geq 3$, $\delta = \varepsilon d^{-s/2}$. Now by the discrete version of Bihari-LaSalle inequality, recalled in Lemma C.10, we have with probability at least $1 - p_{\delta,M}(T)$,

$$m_T - \frac{b}{\sqrt{d}} \geq \frac{b}{\sqrt{d}} \left(1 - \delta\frac{C(s-2)}{2}\left(\frac{b}{\sqrt{d}}\right)^{s-2} T\right)^{-\frac{1}{s-2}}.$$

And as the right hand side is larger than $1/2 + b/\sqrt{d}$ whenever,

$$\delta T \geq \frac{d^{(s-2)/2}}{C(s-2)b^{s-2}},$$

for $d$ large enough compare to $b$. Then taking such a $T$, with probability at least $1 - p_{\delta,M}(T)$, the hitting time is upper bounded by

$$\tau_{1/2}^+ \leq \frac{1}{Cb^{s-2}}\frac{d^{\frac{s-2}{2}}}{\delta}.$$

Now, taking $\delta = \varepsilon d^{-s/2}$, we can check that for $\varepsilon$ small enough, $dT\delta^2 \leq \frac{\varepsilon}{C(s-2)b^{s-2}} = \varepsilon\mathcal{O}(1)$ so that we have that with probability at least $1 - \mathsf{K}\varepsilon$, we have

$$\tau_{1/2}^+ \leq \frac{\mathsf{K}}{\varepsilon} d^{s-1}.$$

### C.3.2  Technical intermediate result to lower bound each term of Eq. (28)

**Lemma C.2** (ODE term). *Conditioned to the event* $\{T \leq \tau_{1/2}^+ \wedge \tau_{2b/\sqrt{d}}^-\}$*, we have the inequality*

$$\sum_{t=0}^{T-1}(1 - m_t)(m_t - b/\sqrt{d})^{s-1} \geq \frac{1}{2^s}\sum_{t=0}^{T-1}m_t^{s-1}.$$

*Proof.* This simply results from the fact that for all $t \leq T - 1$, we have $\{t \leq \tau_{1/2}^+ \wedge \tau_{2b/\sqrt{d}}^-\} \subset \{T \leq \tau_{1/2}^+ \wedge \tau_{2b/\sqrt{d}}^-\}$, so that we can use the inequalities $1 - m \geq 1/2$ and $m - b/\sqrt{d} \geq m/2$. Summing these terms until $T - 1$ gives the proof of the lemma. $\blacksquare$

**Lemma C.3** (First martingale term). *For all* $\lambda > 0$*, we have that*

$$\mathbb{P}\left(\sup_{t \leq T} \delta \left|\sum_{k=0}^{t-1} \nabla^S M_k \cdot \theta^*\right| \geq \lambda\right) \leq \frac{KT\delta^2}{\lambda^2}, \tag{29}$$

*where* $K > 0$*, that depends solely on the model through* $f, \nu$*.*

*Proof.* This is a consequence of Doob's maximal inequality for (sub)martingale. Indeed, for $t \leq T$, let $H_{t-1} = \sum_{k=0}^{t-1} \nabla^S M_k \cdot \theta^*$. We have that $H_t$ is a $\mathcal{F}_t$-adapted martingale and we have the following

upper bounded on its variance:

$$\mathbb{E}[H_{t-1}^2] = \mathbb{E}\left[\left(\sum_{k=0}^{t-1}\nabla^S M_k \cdot \theta^*\right)^2\right]$$

$$= \mathbb{E}\left[\sum_{k=0}^{t-1}\left(\nabla^S M_k \cdot \theta^*\right)^2\right]$$

$$\leq t \sup_{\theta}\mathbb{E}_x\left[\left(\nabla^S M_k \cdot \theta^*\right)^2\right]$$

$$\leq Kt,$$

where the last inequality comes from the Lemma C.8. Now, thanks to Doob's maximal inequality, we have for all $\lambda > 0$,

$$\mathbb{P}\left(\sup_{t \leq T}\delta|H_{t-1}| \geq \lambda\right) \leq \frac{\mathbb{E}[H_{T-1}^2]\delta^2}{\lambda^2} \leq \frac{KT\delta^2}{\lambda^2},$$

and this concludes the proof of the lemma. ∎

**Lemma C.4** (Submartingale term). *For all $\lambda > 0$, if for all $t \leq T$, $m_t \in [2b/\sqrt{d}, 1/2]$, and $\delta$ is such that $\delta \leq \varepsilon/d$, with a small enough constant $\varepsilon > 0$, we have that*

$$\mathbb{P}\left(\delta\sum_{t=0}^{T-1}D_t \leq -\lambda\right) \leq \exp\left(-\frac{\lambda^2}{2K^2\delta^2 T + \lambda\delta(C + \delta\mathsf{M})}\right) \tag{30}$$

*where $K > 0$, that depends solely on the model through $f, \nu$.*

*Proof.* First, recall that we have defined $D_t = \frac{C}{2}(1 - m_t)(m_t - b/\sqrt{d})^{s-1} - \delta|m_t|\,|\nabla_\theta l_t|^2\,\mathbb{1}_{\{|\nabla_\theta l_t|^2 \leq \mathsf{M}\}}$. Let us notice that if $m_t \in [2b/\sqrt{d}, 1/2]$, then $1 - m_t \geq 1/2$ and $(m_t - b/\sqrt{d})^{s-1} \geq m_t^{s-1}/2^{s-1}$. Hence, if $m_t$ lies in such an interval,

$$D_t \geq \frac{C}{2^{s+1}}m_t^{s-1} - \delta|m_t|\,|\nabla_\theta l_t|^2\,\mathbb{1}_{\{|\nabla_\theta l_t|^2 \leq \mathsf{M}\}}$$

$$\geq \frac{C}{2^{s+1}}m_t^{s-1}\left(1 - 2^{s+1}\delta\frac{|\nabla_\theta l_t|^2\,\mathbb{1}_{\{|\nabla_\theta l_t|^2 \leq \mathsf{M}\}}}{Cm_t^{s-2}}\right).$$

Now, for $\delta$ such that $\mathbb{E}\left[1 - 2^{s+1}\delta\frac{|\nabla_\theta l_t|^2\,\mathbb{1}_{\{|\nabla_\theta l_t|^2 \leq \mathsf{M}\}}}{Cm_t^{s-2}}\,\Big|\,\mathcal{F}_{t-1}\right] \geq 0$, $\left(\sum_{k=1}^t D_k\right)_{t \geq 0}$ is a submartingale, which is true as soon as

$$\delta \leq \frac{Cm_t^{s-2}}{2^{s+1}\sup_\theta \mathbb{E}\left[|\nabla_\theta l_t|^2\,\mathbb{1}_{\{|\nabla_\theta l_t|^2 \leq \mathsf{M}\}}\,\Big|\,\mathcal{F}_{t-1}\right]},$$

which is itself true if

$$\delta \leq \frac{C}{4^s\sup_\theta \mathbb{E}\left[|\nabla_\theta l_t|^2\right]},$$

which is implied by the condition required in the lemma given the upper bound on $\mathbb{E}[|\nabla_\theta l_t|^2]$ provided in Lemma C.8. In order to apply Freedman tail inequality for this submartingale, let us provide upper bound on the increments as well as their variance. Indeed, we have, for all $t \geq 0$,

$$|D_t| \leq \frac{C|1 - m_t||m_t - b\sqrt{d}|^{s-1}}{2} + \delta|m_t|\,|\nabla_\theta l_t|^2\,\mathbb{1}_{\{|\nabla_\theta l_t|^2 \leq \mathsf{M}\}}$$

$$\leq \frac{C + \delta\mathsf{M}}{2},$$

and in virtue of the inequality $(a+b)^2 \le 2(a^2+b^2)$, we have

$$
\mathbb{E}\left[D_t^2 \,|\, \mathcal{F}_{t-1}\right] \le 2\left(\frac{C^2|1-m_t|^2|m_t - b\sqrt{d}|^{2(s-1)}}{4} + \delta^2|m_t|^2 \mathbb{E}\left[|\nabla_\theta l_t|^4 \mathbb{1}_{\{|\nabla_\theta l_t|^2 \le \mathsf{M}\}}\right]\right)
$$

$$
\le \frac{C^2 + \delta^2 \mathbb{E}\left[|\nabla l|^4\right]}{2}
$$

$$
\le \frac{C^2 + K\delta^2 d^2}{2}
$$

$$
\le K^2.
$$

Hence, by the Freedman tail inequality recalled in Theorem C.9, for all $\lambda > 0$,

$$
\mathbb{P}\left(\delta \sum_{t=0}^{T-1} D_t \le -\lambda\right) \le \exp\left(-\frac{\lambda^2}{2K^2\delta^2 T + \lambda\delta(C + \delta\mathsf{M})}\right),
$$

which concludes the proof of the Lemma. ∎

**Lemma C.5** (First discretization term). *We have that, almost surely*

$$
\mathbb{P}\left(\sup_{t \le T} \delta^2 \sum_{t=0}^{T-1} |m_t|\,|\nabla_\theta l_t|^2 \mathbb{1}_{\{|\nabla_\theta l_t|^2 > \mathsf{M}\}} \ge \lambda\right) \le \frac{KT\delta^2 d^2}{\lambda\mathsf{M}}, \tag{31}
$$

*where $K > 0$ depends solely on the model through $f, \nu$.*

*Proof.* This term is handled via a combination of Markov and Cauchy-Schwartz inequalities. First, notice that,

$$
\sup_{t \le T} \delta^2 \sum_{t=0}^{T-1} |m_t|\,|\nabla_\theta l_t|^2 \mathbb{1}_{\{|\nabla_\theta l_t|^2 > \mathsf{M}\}} \le T\delta^2 \sup_{t \le T}\left\{|m_t|\,|\nabla_\theta l_t|^2 \mathbb{1}_{\{|\nabla_\theta l_t|^2 > \mathsf{M}\}}\right\}.
$$

Furthermore, for all $t \le T$, all $\lambda > 0$, via Markov inequality, then Cauchy-Schwartz inequality,

$$
\mathbb{P}\left(|m_t|\,|\nabla_\theta l_t|^2 \mathbb{1}_{\{|\nabla_\theta l_t|^2 > \mathsf{M}\}} \ge \lambda\right) \le \frac{\mathbb{E}\left[|m_t|\,|\nabla_\theta l_t|^2 \mathbb{1}_{\{|\nabla_\theta l_t|^2 > \mathsf{M}\}}\right]}{\lambda}
$$

$$
\le \frac{\sqrt{\mathbb{E}\left[|\nabla_\theta l_t|^4\right]}\sqrt{\mathbb{P}\left(|\nabla_\theta l_t|^2 > \mathsf{M}\right)}}{\lambda}
$$

$$
\le \frac{\sqrt{\mathbb{E}\left[|\nabla_\theta l_t|^4\right]}\sqrt{\mathbb{E}\left[|\nabla_\theta l_t|^4\right]/\mathsf{M}^2}}{\lambda}
$$

$$
\le \frac{\mathbb{E}\left[|\nabla_\theta l_t|^4\right]}{\lambda\mathsf{M}}
$$

$$
\le \frac{Kd^2}{\lambda\mathsf{M}},
$$

where the last inequality is due to Lemma C.8. Multiplying this bound by $T\delta^2$ ends the proof the lemma. ∎

**Lemma C.6** (Second discretization term). *For all $\lambda > 0$, we have that*

$$
\mathbb{P}\left(\sup_{t \le T} \delta^3 \sum_{k=0}^{t-1} \xi_k \ge \lambda\right) \le \frac{KTd\delta^3}{\lambda}, \tag{32}
$$

*where $K > 0$ depends solely on the model through $f, \nu$.*

*Proof.* Recall that $\xi_k = |\nabla_\theta l(\theta_k, x_k)|^2 |\nabla^S l(\theta_k, x_k) \cdot \theta^*|$. The bound follows from an application of Markov's inequality. Indeed, since all the terms of the sum are positive, the supremum is attained in $t = T - 1$, and we shall only consider this case. For $\lambda > 0$,

$$\mathbb{P}\left(\delta^3 \sum_{t=0}^{T-1} \xi_t \geq \lambda\right) \leq \frac{\delta^3}{\lambda} \mathbb{E}\left[\sum_{t=0}^{T-1} \xi_t\right]$$

$$\leq \frac{T\delta^3}{\lambda} \sup_\theta \left\{ \mathbb{E}_x[|\nabla_\theta l(\theta, x)|^2 |\nabla^S l(\theta, x) \cdot \theta^*|] \right\}$$

$$\leq \frac{T\delta^3}{\lambda} \sup_\theta \left\{ \sqrt{\mathbb{E}_x\left[|\nabla_\theta l(\theta, x)|^4\right]} \sqrt{\mathbb{E}_x\left[|\nabla^S l(\theta, x) \cdot \theta^*|^2\right]} \right\}$$

$$\leq \frac{T\delta^3}{\lambda} \sqrt{\sup_\theta \mathbb{E}_x\left[|\nabla_\theta l(\theta, x)|^4\right]} \sqrt{\sup_\theta \mathbb{E}_x\left[|\nabla^S l(\theta, x) \cdot \theta^*|^2\right]}$$

$$\leq \frac{T\delta^3}{\lambda} \sqrt{\sup_\theta \mathbb{E}_x\left[|\nabla_\theta l(\theta, x)|^4\right]} \sqrt{\sup_\theta \mathbb{E}_x\left[|\nabla^S l(\theta, x) \cdot \theta^*|^2\right]}$$

$$\leq \frac{T\delta^3}{\lambda} \sqrt{Kd^2}\sqrt{K}$$

$$\leq \frac{KTd\delta^3}{\lambda},$$

where the penultimate inequality comes from Lemma C.8. ∎

## C.4 Strong recovery

The reasoning is almost identical to the one of the previous section, except from the fact that instead of tracking the growing movement on $(m_t)_{t \geq 0}$, we will track the decaying movement of $(1 - m_t)_{t \geq 0}$.

### C.4.1 Upper bound on the residual

As said in the main text, we place ourselves *after* the weak recovery time. Thanks to the Markovian property of the SGD dynamics, we have the equality between all time $s > 0$ marginal laws of

$$\left(\theta_{\tau_{1/2}^+ + s} \,\middle|\, \tau_{1/2}^+, \theta_{\tau_{1/2}^+}\right) \overset{\text{Law}}{=} \left(\theta_s \,\middle|\, \theta_s = \theta_{\tau_{1/2}^+}\right),$$

and hence the strong recovery question is equivalent to study the dynamics with initialization such that $m_\theta = 1/2$. As demonstrated before we have that $\mathbb{P}(\tau_{1/2}^+ < \infty) \geq 1 - \mathsf{K}\varepsilon$ so that up to $\varepsilon$ terms, this conditioning does not hurt the probability of the later events. In fact this conditioning seems even artificial as it seems provable that $\tau_{1/2}^+$ is almost surely finite. Yet, we leave this more precise study for another time.

### C.4.2 A (slightly) different decomposition

Let us define for all $t \in \mathbb{N}$, the residual $u_t = 1 - m_{t + \tau_{1/2}^+} > 0$, and thanks to the lower bound given by Eq. (27), we have

$$u_{t+1} \leq u_t - C\delta u_t(m_t - b/\sqrt{d})^{s-1} + \delta \nabla^S M_t \cdot \theta^* + \delta^2 |m_t| |\nabla l(x_t, \theta_t)|^2 + \delta^3 \xi_t,$$

From there, the proof is similar to the weak recovery case, except that the extra-care we used for the term $\delta^2 |m_t| |\nabla l(x_t, \theta_t)|^2$ is not necessary. We use simply the decomposition of this term in a second martingale term

$$N_t = |m_t| |\nabla l(x_t, \theta_t)|^2 - \mathbb{E}\left[|m_t| |\nabla l(x_t, \theta_t)|^2 | \mathcal{F}_{t-1}\right]$$

and the drift that we directly upper bound as $\mathbb{E}\left[|m_t| |\nabla l(x_t, \theta_t)|^2 | \mathcal{F}_{t-1}\right] \leq Kd$. Now similarly to Lemma C.3, we have the upper bound:

**Lemma C.7** (New martingale term). *For all $\lambda > 0$, we have that*

$$\mathbb{P}\left(\sup_{t \leq T} \delta^2 \left|\sum_{k=0}^{t-1} N_k\right| \geq \lambda\right) \leq \frac{Kd^2 T\delta^4}{\lambda^2}, \tag{33}$$

*where $K > 0$, that depends solely on the model through $f, \nu$.*

*Proof.* This is a consequence of Doob's maximal inequality for the martingale. Indeed, for $t \leq T$, let $H_{t-1} = \sum_{k=0}^{t-1} N_k$. We have that $N_t$ is a $\mathcal{F}_t$-adapted martingale and we have the following upper bounded on its variance:

$$\mathbb{E}[N_{t-1}^2] = \mathbb{E}\left[\left(\sum_{k=0}^{t-1} \nabla^S M_k \cdot \theta^*\right)^2\right]$$

$$= \mathbb{E}\left[\sum_{k=0}^{t-1} N_k^2\right]$$

$$\leq t \sup_\theta \mathbb{E}_x (N_k)^2$$

$$\leq Kd^2 t,$$

where the last inequality comes from the Lemma C.8. Now, thanks to Doob's maximal inequality, we have for all $\lambda > 0$,

$$\mathbb{P}\left(\sup_{t \leq T} \delta^2 |H_{t-1}| \geq \lambda\right) \leq \frac{\mathbb{E}[H_{T-1}^2]\delta^4}{\lambda^2} \leq \frac{Kd^2 T\delta^4}{\lambda^2},$$

and this concludes the proof of the lemma. ∎

Now, everything is in order to prove the Theorem 3.3.

### C.4.3 Proof of Theorem 3.3

Let us fix a small number $\varepsilon > 0$. As previously, thanks to Lemmas C.3, C.6, C.7, there exists $K > 0$ that depends solely on the model such that we have the following upper bound: for all $\lambda$, and $t \leq \tau_{1/3}^- \wedge \tau_{1-\varepsilon}^+$ summing between times 0 and $t$,

$$u_t \leq u_0 - \frac{C\delta}{4^{s-1}} \sum_{k=0}^{t-1} u_k + K\delta^2 d + 3\lambda,$$

with probability larger that $1 - \left(\frac{Kt\delta^2}{\lambda^2} + \frac{Kd^2 t\delta^4}{\lambda^2} + \frac{Ktd\delta^3}{\lambda}\right)$ and $d$ large enough. Let us choose $\lambda = 1/16$ and $\delta$ small enough so that $K\delta^2 d \leq \lambda$. Hence, realizing that $u_0 \leq 1/2$, we have

$$u_t \leq \frac{3}{4} - \frac{C\delta}{4^{s-1}} \sum_{k=0}^{t-1} u_k ,$$

with probability at least $1 - Kt\delta^2(1 + d^2\delta^2 + d\delta) \gtrsim 1 - Kt\delta^2$, as we choose in any case $\delta = \varepsilon\mathcal{O}(1)$. Note that we used the same convention as in the weak recovery case that K denotes *any* constant that simply depend on the model. We have by Grönwall inequality (Lemma C.10)

$$u_t \leq \frac{3}{4}\left(1 - \frac{C\delta}{4^{s-1}}\right)^t \leq \frac{3}{4}e^{-\frac{C\delta}{4^{s-1}}t}.$$

Hence, as the right end side is smaller than $\varepsilon$ for the time

$$t\delta \geq \frac{4^{s-1}}{C}\log(1/\varepsilon),$$

we choose such a $t$, so that with probability at least $1 - K\delta\log(1/\varepsilon)$, the delayed hitting time $\overline{\tau}_{1-\varepsilon}^+ := \inf\{t \geq 0, u_t \leq \varepsilon\}$ satisfies

$$\overline{\tau}_{1-\varepsilon}^+ \leq \frac{4^{s-1}}{C\delta}\log(1/\varepsilon),$$

and taking $\delta = \varepsilon/d$ gives that with a probability at least $1 - K\varepsilon\log(1/\varepsilon)/d$, we have

$$\overline{\tau}_{1-\varepsilon}^+ \leq \frac{4^{s-1}}{C\varepsilon}d\log(1/\varepsilon).$$

Considering that $d$ is large and $\varepsilon$ is simply a constant we get that $1 - K\varepsilon\log(1/\varepsilon)/d \geq 1 - K\varepsilon$ and and this concludes the proof of Theorem 3.3.

## C.5 Some technical bounds

We end this section by providing (i) some necessary technical technical bound on the quantities appearing in the SGD controls (ii) some discrete versions of Grönwall-type lemmas.

### C.5.1 Technical bounds on models expectations

**Lemma C.8** (Technical bounds). *We have that there exists a constant $K > 0$ solely depending on the function $\phi$ and the distribution $\nu$ such that:*

$$\sup_{\theta \in \mathcal{S}_{d-1}} \mathbb{E}_x \left[ \langle \nabla_\theta^{\mathcal{S}} M(x, \theta), \theta_* \rangle^2 \right] \leq K , \quad and \quad \sup_{\theta} \mathbb{E}_x \left[ |\nabla^S l(\theta, x) \cdot \theta^*|^2] \right] \leq K \tag{34}$$

$$\sup_{\theta \in \mathcal{S}_{d-1}} \mathbb{E}_x[|\nabla_\theta l(\theta, x)|^2] \leq Kd, \tag{35}$$

$$\sup_{\theta \in \mathcal{S}_{d-1}} \mathbb{E}_x[|\nabla_\theta l(\theta, x)|^4] \leq Kd^2. \tag{36}$$

*Proof.* In all the following proof we consider any $\theta \in \mathcal{S}_{d-1}$. Notice that we have the following calculation that is common to all the bounds we cover

$$\nabla l(\theta, x) = x \phi'(x \cdot \theta) \phi(x \cdot \theta_*)$$

We treat the three bounds separately.

*First terms.* We have that for all $x \in \mathbb{R}^d$,

$$M(x, \theta) = l(x, \theta) - \mathbb{E}_\nu[l(x, \theta)],$$

hence

$$\begin{aligned}
\nabla_\theta^{\mathcal{S}} M(x, \theta) &= \nabla_\theta^{\mathcal{S}} l(x, \theta) - \mathbb{E}_\nu[\nabla_\theta^{\mathcal{S}} l(x, \theta)] \\
&= \nabla_\theta l(x, \theta) - \mathbb{E}_\nu[\nabla_\theta l(x, \theta)] - (\theta \cdot \nabla_\theta l(x, \theta))\theta + \mathbb{E}_\nu[(\theta \cdot \nabla_\theta l(x, \theta))\theta],
\end{aligned}$$

and finally,

$$\nabla_\theta^{\mathcal{S}} M(x, \theta) \cdot \theta_* = \nabla_\theta l(x, \theta) \cdot \theta_* - \mathbb{E}_\nu[\nabla_\theta l(x, \theta) \cdot \theta_*] - (\theta \cdot \nabla_\theta l(x, \theta))m + \mathbb{E}_\nu[(\theta \cdot \nabla_\theta l(x, \theta))m].$$

hence thanks to applying the inequality $(a + b)^2 \leq 2a^2 + 2b^2$, this amounts to bound first

$$\mathbb{E}_x \left( \nabla_\theta l(x, \theta) \cdot \theta_* \right)^2 = \mathbb{E}_x \left[ (x \cdot \theta_*)^2 \phi'^2(x \cdot \theta) \phi^2(x \cdot \theta_*) \right] \leq K,$$

and second

$$\mathbb{E}_x \left( (\nabla_\theta l(x, \theta) \cdot \theta)m \right)^2 \leq \mathbb{E}_x \left[ (x \cdot \theta)^2 \phi'^2(x \cdot \theta) \phi^2(x \cdot \theta_*) \right] \leq K.$$

*Second term.* We have

$$\mathbb{E}_x \left| \nabla_\theta l(x, \theta) \right|^2 = \mathbb{E}_x \left[ |x|^2 \phi'^2(x \cdot \theta) \phi^2(x \cdot \theta_*) \right] \leq Kd .$$

*Third term.* We have similarly

$$\mathbb{E}_x \left| \nabla_\theta l(x, \theta) \right|^4 = \mathbb{E}_x \left[ |x|^4 \phi'^2(x \cdot \theta) \phi^2(x \cdot \theta_*) \right] \leq Kd^2 .$$

∎

### C.5.2 Standard tail probabilities for submartingales

We recall here a theorem on submartingales from Freedman. This is an adaptation from Theorem 4.1 stated in Freedman [1975].

**Theorem C.9** (Submartingale tail bound). *Suppose that $(X_t)_{t \in \mathbb{N}}$ is random sequence adapted to a filtration $(\mathcal{F}_t)_{t \in \mathbb{N}}$. For $T \geq 1$, suppose there exist $a, b > 0$ such that $\mathbb{E}[X_t \mid \mathcal{F}_{t-1}] \geq 0$, the almost sure upper-bound $\sup_{t \leq T} |X_t| \leq a$ as well as $\sup_{t \leq T} \mathbb{E}[X_t^2 \mid \mathcal{F}_{t-1}] \leq b$, then for all $\lambda > 0$,*

$$\mathbb{P} \left( \sum_{k=1}^T X_k \leq -\lambda \right) \leq \exp \left( -\frac{\lambda^2}{2(Tb + \lambda a)} \right) \tag{37}$$

### C.5.3 Discrete Grönwall and Bihari-Lasalle bounds

We now turn to stating a classical comparison lemma for recursive inequalities.

**Lemma C.10** (Grönwall and Bihari-Lasalle). *We have the bounds for the recursive inequalities:*

***Case*** $s = 2$. *Suppose* $(m_t)_{t\in\mathbb{N}}$ *satisfies for* $s \geq 3$, *and positives numbers* $a, b > 0$, *and* $b < a/2 \wedge 1$,

$$m_t \geq a + b\sum_{k=0}^{t-1} m_k, \qquad then, \quad m_t \geq a\left(1+b\right)^t \tag{38}$$

$$m_t \leq a - b\sum_{k=0}^{t-1} m_k, \qquad then, \quad m_t \leq a\left(1-b\right)^t. \tag{39}$$

***Case*** $s \geq 3$. *Suppose* $(m_t)_{t\in\mathbb{N}}$ *satisfies for* $s \geq 3$, *and positives numbers* $a, b > 0$:

$$m_t \geq a + b\sum_{k=0}^{t-1} m_k^{s-1}, \qquad then, \quad m_t \geq a\left(1 - (s-2)ba^{s-2}t\right)^{-\frac{1}{s-2}}. \tag{40}$$

*Proof.* The case $s = 2$ is known to be the discrete version of the Grönwall lemma and is treated in all standard textbooks, the case $s \geq 3$ referred to as the Bihari-Lasalle inequality is for example proven in Appendix C of Arous et al. [2021]. ∎

## D The LPG property in the symmetric case: proofs of Section 4.1

### D.1 Useful Facts about Gegenbauer Polynomials

We recall known facts on Gegenbauer Polynomials.

**Definitions.** Recall that $P_{j,d}$ denotes the Gegenbauer polynomial of degree $j$ and dimension $d$, normalized so that $P_{j,d}(1) = 1$ for all $j, d$. We denote also $\bar{P}_{j,\lambda}$ the Gegenbauer polynomials normalized so that $\|\bar{P}_{j,\lambda}\|^2_{L^2(\mathbb{R}, u_{2\lambda+2})} = \pi 2^{1-2\lambda}\frac{\Gamma(j+2\lambda)}{(j+\lambda)\Gamma^2(\lambda)\Gamma(j+1)}$. Throughout the proof, we will use either $d$, and from time to time the mute symbol $\lambda$ to denote the dimension variable of Gegenbauer polynomials. They satisfy the following recurrence:

$$(j+1)\bar{P}_{j+1,\lambda}(t) = 2(j+\lambda)t\bar{P}_{j,\lambda}(t) - (j+2\lambda-1)\bar{P}_{j-1,\lambda}(t), \tag{41}$$

with first terms: $\bar{P}_{0,\lambda}(t) = 1$ and $\bar{P}_{1,\lambda}(t) = 2\lambda t$.

**Rodrigues Formula for Gegenbauer Polynomials.** The Gegenbauer polynomials can be represented as repeated derivatives of a simple polynome.

**Proposition D.1** ([Frye and Efthimiou, 2012, Proposition 4.19]). *We have the formula*

$$P_{j,d}(t) = \frac{(-1)^j}{2^j(j+(d-3)/2)_j}(1-t^2)^{(3-d)/2}\left(\frac{d}{dt}\right)^j(1-t^2)^{j+(d-3)/2}, \tag{42}$$

*where* $(x)_j = \prod_{k=0}^{j-1}(x-k)$ *is the falling factorial.*

**Hecke-Funk Formula.** Recall that we use the notation $\tau_d$ to denote the uniform distribution on the sphere and $u_d$ the distribution of, e.g., its first coordinate: $u_d \propto (1-t^2)^{(d-3)/2}\mathbb{1}_{[-1,1]}$.

**Theorem D.2** ([Frye and Efthimiou, 2012, Theorem 4.24]). *For* $\theta, \theta' \in \mathcal{S}_{d-1}$, $f \in L^2_{u_d}(\mathbb{R})$ *and* $j \in \mathbb{N}$,

$$\langle f_\theta, (P_{j,d})_{\theta'}\rangle_{\tau_d} = \Omega_{d-2}P_{j,d}(\theta\cdot\theta')\langle f, P_{j,d}\rangle_{u_d}$$

$$= \Omega_{d-2}P_{j,d}(\theta\cdot\theta')\int_{-1}^{1}f(t)P_{j,d}(t)(1-t^2)^{(d-3)/2}dt. \tag{43}$$

**Fact D.3.** *[Derivative Representation] We have the following derivation property for all* $j, d$:

$$P'_{j,d} = \frac{j(j+d-2)}{(d-1)}P_{j-1,d+2}. \tag{44}$$

*Proof.* Recall the normalization relationships $\lambda = \frac{d}{2} - 1$, $\bar{P}_{j,\lambda}(1) = \frac{\Gamma(j+2\lambda)}{\Gamma(j+1)\Gamma(2\lambda)}$ , $P_{j,d} = \frac{\Gamma(j+1)\Gamma(d-2)}{\Gamma(j+d-2)}\bar{P}_{j,\frac{d}{2}-1}$, as well as the identity $\bar{P}'_{j,\lambda} = 2\lambda\bar{P}_{j-1,\lambda+1}$. Thus,

$$
\begin{aligned}
P'_{j,d} &= \frac{\Gamma(j+1)\Gamma(d-2)}{\Gamma(j+d-2)}\bar{P}'_{j,\frac{d}{2}-1} \\
&= 2\frac{\Gamma(j+1)\Gamma(d-2)}{\Gamma(j+d-2)}(\frac{d}{2}-1)\bar{P}_{j-1,\frac{d}{2}} \\
&= (d-2)\frac{\Gamma(j+1)\Gamma(d-2)}{\Gamma(j+d-2)}\frac{\Gamma(j-1+d)}{\Gamma(j)\Gamma(d)}P_{j-1,d+2} \\
&= \frac{j(j+d-2)}{(d-1)}P_{j-1,d+2}
\end{aligned}
$$

(45)

■

We have the following bound of the location of the largest root $z_{j,d}$ of $P_{j,d}$:

**Fact D.4** (Bound on the Largest Root,[Area et al., 2004, Corollary 2.3]).

$$
z_{j,d} \leq \sqrt{\frac{(j-1)(j+d-4)}{(j+d/2-3)(j+d/2-2)}} \cos(\pi/(j+1)) .
$$

(46)

And we have the following bound on the Taylor expansion of the Gegenbauer polynomials:

**Fact D.5** (Taylor Upper bound beyond largest root).

$$
P_{j,d}(t) \geq (t-z_{j,d})^j , \text{ for } t \geq z_{j,d} ,
$$

*Proof.* Note that all families of orthogonal polynomials have exclusively real, simple roots. Therefore, by Rolle's theorem, the $j-1$ critical points of $P_{j,d}$ must be interlaced with the $j$ zeroes. So all zeroes of $P'_{j,d}$ are upper bounded by $z_{j,d}$. Futhermore, by Fact D.3, $P'_{j,d}$ is itself an orthogonal polynomial. So applying this argument recursively, we see the zeros of $P^{(k)}_{j,d}$ for $k \leq j$ are all upper bounded by $z_{j,d}$.

Note also by Fact D.3 that, because $P_{j,d}(1) = 1$ for any choice of $j$ and $d$, it follows that $P^{(k)}_{j,d}(1) > 0$. This implies $P^{(k)}_{j,d}(z_{j,d}) > 0$, as in order to flip signs there would need to be a zero in the range $[z_{j,d}, 1]$ which we've confirmed above cannot exist.

Now, consider a Taylor expansion

$$
P_{j,d}(t) = \sum_{i=0}^{j} c_i(t-z_{j,d})^i
$$

(47)

Observe that $P^{(k)}_{j,d}(z_{j,d}) = k!c_k$, and therefore by the above argument we have $c_k > 0$. So it remains to show that $c_j \geq 1$.

Consider applying Fact D.3 repeatedly, then we have:

$$
P^{(j)}_{j,d}(1) = \frac{j! \prod_{l=1}^{j}(j+d-3+l)}{\prod_{l=1}^{j}(d-3+2l)}
$$

(48)

$$
= j! \prod_{l=1}^{j} \frac{j+d-3+l}{d-3+2l}
$$

(49)

$$
\geq j!
$$

(50)

And from the fact that $P^{(j)}_{j,d}(1) = j!c_j$, we conclude $c_j \geq 1$. ■

## D.2 Proof of Proposition 4.1

**Proposition D.6** (Loss representation, restated)**.** *The $\beta_{j,d}$ defined in (14) have the integral representation*

$$\beta_{j,d} = \langle \phi, \mathcal{K}_j \phi \rangle_{L^2(\mathbb{R},\eta)} , \tag{51}$$

*where $\mathcal{K}_j$ is a positive semi-definite integral operator of $L^2_\eta$ that depend solely on $\rho$ and $\phi$, with kernel*

$$\mathcal{K}_j(t,t') = \frac{\Omega_{d-2} N(j,d)}{\Omega_{d-1}} \int_0^\infty P_j(r^{-1}t) P_j(r^{-1}t') \bar{u}_d(r^{-1}t) \bar{u}_d(r^{-1}t') \rho(dr) , \tag{52}$$

*where we defined the conditional density*

$$\bar{u}_d(r^{-1}t) = \frac{r^{-1} u_d(r^{-1}t)}{\int_0^\infty (r')^{-1} u_d((r')^{-1}t) \rho(dr')} .$$

*Moreover, we have*

$$\mathbb{E}_\eta[\phi^2] = \frac{\Omega_{d-2}}{\Omega_{d-1}} \sum_j \beta_{j,d} = \frac{\Gamma((d-2)/2)}{\sqrt{\pi}\Gamma((d-1)/2)} \sum_j \beta_{j,d} . \tag{53}$$

*Proof.* The marginal conditioned on $\|x\| = r$ is precisely given by $\eta(x_1 = t \mid \|x\| = r) = r^{-1} u_d(r^{-1}t)$, so

$$\eta(t) = \int_0^\infty r^{-1} u_d(r^{-1}t) \rho(dr) .$$

We have

$$\alpha_{j,r} = \|P_j\|^{-2} \int_{-1}^1 P_j(t) \phi^{(r)}(t) \tau_d(dt) = \|P_j\|^{-2} \int_{-1}^1 P_j(t) \phi(rt) u_d(t) dt$$

$$= \frac{\Omega_{d-2} N(j,d)}{\Omega_{d-1}} r^{-1} \int_{-\infty}^\infty \phi(t) P_j(r^{-1}t)(1 - r^{-2}t^2)_+^{(d-3)/2} dt , \tag{54}$$

so

$$\beta_{j,d} = \frac{\Omega_{d-2} N(j,d)}{\Omega_{d-1}} \int_0^\infty r^{-2} \iint_{-\infty}^\infty \phi(t) P_j(r^{-1}t)(1 - r^{-2}t^2)_+^{(d-3)/2} \phi(t') P_j(r^{-1}t')(1 - r^{-2}(t')^2)_+^{(d-3)/2} dt dt' \rho(dr)$$

$$= \langle \phi, \mathcal{K}_j \phi \rangle_{L^2(\mathbb{R},\eta)} , \tag{55}$$

with the $L^2(\mathbb{R},\eta)$ positive semi-definite integral kernel operator

$$\mathcal{K}_j(t,t') = \frac{\Omega_{d-2} N(j,d)}{\Omega_{d-1}} \eta(t)^{-1} \eta(t')^{-1} \int_0^\infty r^{-2} P_j(r^{-1}t)(1 - r^{-2}t^2)_+^{(d-3)/2} P_j(r^{-1}t')(1 - r^{-2}(t')^2)_+^{(d-3)/2} \rho(dr)$$

$$= \frac{\Omega_{d-2} N(j,d)}{\Omega_{d-1}} \int_0^\infty P_j(r^{-1}t) P_j(r^{-1}t') \bar{u}_d(r^{-1}t) \bar{u}_d(r^{-1}t') \rho(dr) , \tag{56}$$

where we defined the conditional density

$$\bar{u}_d(r^{-1}t) = \frac{r^{-1} u_d(r^{-1}t)}{\int_0^\infty (r')^{-1} u_d((r')^{-1}t) \rho(dr')} .$$

Finally, let us establish (57). We have

$$\mathbb{E}_\eta \phi^2 = \mathbb{E}_\rho[\mathbb{E}_{x_1 | \|x\| = r} \mathbb{E}\phi(x_1)^2]$$

$$= \mathbb{E}_\rho[\mathbb{E}_{u_d}(\phi^{(r)})^2]$$

$$= \mathbb{E}_\rho \sum_j \alpha_{j,d,r}^2 \|P_j\|^2$$

$$= \mathbb{E}_\rho \sum_j \alpha_{j,d,r}^2 \frac{\Omega_{d-1}}{\Omega_{d-2} N(j,d)}$$

$$= \frac{\Omega_{d-1}}{\Omega_{d-2}} \mathbb{E}_\rho \sum_j \bar{\alpha}_{j,r,d}^2 = \frac{\Omega_{d-1}}{\Omega_{d-2}} \sum_j \beta_{j,d} . \tag{57}$$

∎

### D.3 Proof of Proposition 4.2

*Proof.* If $\beta_{j,d} = 0$ for $j < s$, then $\alpha_{j,r,d} = 0$ for $j < s$ and $\rho$-ae $r$. We want to show that for any polynomial $Q$ of degree $j' < s$, we must have $\langle \phi, Q \rangle_\eta = 0$.

For each $r$, consider $Q^{(r)}(t) = Q(rt)$, which is also a polynomial of degree $j' < s$, and its decomposition as $Q^{(r)} = \sum_{j=0}^{j'} b_{j,j',r} P_{j,d}$, which only involves terms of degree $j' < s$ since Gegenbauer polynomials of degree up to $r$ span all polynomials of degree up to $r$. We have

$$
\begin{aligned}
\langle \phi, Q \rangle_\eta &= \mathbb{E}_\eta[\phi(x) Q(x)] \\
&= \mathbb{E}_\rho \mathbb{E}_{x_1 | \|x\| = r}[\phi(x) Q(x)] \\
&= \mathbb{E}_\rho \mathbb{E}_{u_d}[\phi^{(r)}(x) Q^{(r)}(x)] \\
&= \mathbb{E}_\rho[\sum_{j \leq j'} b_{j,j',r} \alpha_{j,r,d}] = 0 \ .
\end{aligned}
\tag{58}
$$

∎

### D.4 Proof of Proposition 4.3

**Proposition 4.3** (Spectral characterization of LPG). *Suppose there exist constants $K, C > 0$ and $s \in \mathbb{N}$ such that we both have $\beta_{s,d} \geq C$ and $\sum_{j>s} \beta_{j,d} j(j+d-2) v_{j-1,d+2} \leq K d^{(3-s)/2}$. Then, taking $s^*$ as the infimum of such $s$, $L$ has the property $\mathsf{LPG}(s^*-1, z_{s^*,d})$. In particular, whenever $s^* \ll d$, we have $z_{s^*,d} \leq 2\sqrt{s^*/d}$.*

*Proof.* Assume first that there are $\bar{C}, \bar{\zeta}$ such that

$$
P'_{s,d}(t) \geq \bar{C}(t - \bar{\zeta})^{s-1} \ , \quad \text{for } t \geq \bar{\zeta} \ .
\tag{59}
$$

Now, let

$$
B = \frac{1}{d-1} \sum_{j \geq s} \beta_{j,d} j(j+d-2) v_{j-1,d+2} < 1 \ ,
\tag{60}
$$

and define

$$
\zeta^* := \left( \frac{B}{\beta_{s,d} \bar{C}} \right)^{1/(s-1)} + \bar{\zeta} \ .
\tag{61}
$$

From (59), (60) and (61) we verify that $\ell'(m) = \sum_j \beta_{j,d} P'_{j,d}(m)$ satisfies, for $m \geq \zeta^*$,

$$
\ell'(m) \geq \beta_{s,d} \bar{C} \left( (m - \bar{\zeta})^{s-1} - (m - \zeta^*)^{s-1} \right) \geq \beta_{s,d} \bar{C} \left[ \left( \frac{1 - \bar{\zeta}}{1 - \zeta^*} \right)^{s-1} - 1 \right] (m - \zeta^*)^{s-1} \ .
$$

Finally, we have that for any $j, d$, the largest root $z_{j,d}$ satisfies $z_{j,d} \leq \sqrt{\frac{(j-1)(j+2d-2)}{(j+d-2)(j+d-1)}} \simeq j/\sqrt{d}$ and

$$
P_{j,d}(t) \geq \frac{1}{2}(t - z_{j,d})^s \ , \quad \text{for } t \geq z_{j,d} \ ,
$$

which implies that

$$
P'_{s,d}(t) \geq \frac{s(s+d-2)}{2(d-1)}(t - z_{s-1,d+2})^{s-1} \ , \quad \text{for } t \geq z_{s-1,d+2} \ .
\tag{62}
$$

We thus have $\bar{C} = \frac{s(s+d-2)}{2(d-1)}$ with $\bar{\zeta} = z_{s-1,d+2}$.

Finally, we verify that

$$
\frac{d^{(s-1)/2}}{s(s+d-2)} \sum_{j>s} \beta_{j,d} j(j+d-2) v_{j-1,d+2} \leq K
\tag{63}
$$

ensures a local polynomial growth of order $s - 1$ at scale $O(1/\sqrt{d})$. Indeed, plugging (63) into (60), together with $\beta_{s,d} \geq C$ yields

$$
\left( \frac{B}{\beta_{s,d} \bar{C}} \right)^{1/(s-1)} \leq (CK)^{1/(s-1)} d^{-1/2} \ ,
\tag{64}
$$

which shows that $\zeta^* = O(1/\sqrt{d})$. Finally, we observe that $\bar{C} \geq s = \Theta(1)$ if $s < d$.

∎

## D.5  Proof of Theorem 4.5

*Proof.* To prove the theorem, we will establish the sufficient conditions of Proposition 4.3 under our mild assumptions. The key technical results we need are explicit bounds for $v_{j,d}$ and for the sum $\sum_j j^2 \beta_{j,d}$, established in the following two lemmas. Since the parameter $\lambda = d/2 - 1$ is more convenient to express many relationships in Gegenbauer polynomials, we will adopt it in this proof instead of $d$, without loss of generality.

**Lemma D.7** (Control of $v_{j,\lambda}$). *We have*

$$
v_{j,\lambda} \lesssim
\begin{cases}
\left[ 1 - \left( \frac{\lambda}{j+\lambda} \right)^2 \right]^{j/2} & \text{if } j = \Omega(1) , \\
\lambda^{\frac{(\alpha-1)\lambda^\alpha}{2}} & \text{if } j = \Theta(\lambda^\alpha), \text{ with } 0 < \alpha < 1 , \\
e^{-\frac{1}{2}\lambda^{2-\alpha}} & \text{if } j = \Theta(\lambda^\alpha) \text{ with } 1 \le \alpha < 3/2 , \\
e^{-\lambda} & \text{if } j = \Omega(\lambda^{3/2}) .
\end{cases}
\tag{65}
$$

**Lemma D.8** (Decomposition of derivative). *If $\phi \in L^2(\mathbb{R}, \mu)$ is such that $\phi' \in L^4(\mathbb{R}, \eta)$ and $\mathbb{E}_\rho[r^4] < \infty$, then $\beta_j = \langle \phi, \mathcal{K}_j \phi \rangle$ satisfies*

$$
\sum_j j^2 \beta_{j,d} \le \frac{\Omega_{d-2}}{\Omega_{d-1}} \mathbb{E}_\rho[r^4]^{1/2} \|\phi'\|_{L^4(\eta)}^2 = O(1/d) .
\tag{66}
$$

Let $s = \inf\{ j; \beta_{j,d} \ne 0 \}$. We need to verify that there exists a constant $K > 0$ such that

$$
\sum_{j>s} \beta_{j,d} j(j + d - 2) v_{j-1,d+2} \le K d^{(3-s)/2} .
\tag{67}
$$

We will control the LHS by splitting it into appropriate regions, determined by $J_i$, $i \in \{1, 2, 3\}$. Let $\alpha = \frac{4}{1+s}$ and $J_1 = \frac{\lambda^\alpha}{2}$. From Lemma D.7, part (i) we have that $v_{j,\lambda} \le C \left( \frac{j(j+2\lambda)}{(j+\lambda)^2} \right)^{j/2}$, and in particular $v_{j,\lambda} \le C\lambda^{(\alpha-1)j/2}$ for $j \le J_1$. As a result, using Lemma D.8,

$$
\sum_{j=s+1}^{J_1} \beta_j j(j + \lambda) v_{j-1,\lambda+1} \le \lambda^{(\alpha-1)(s+1)/2} \sum_{j=s+1}^{J_1} \beta_j j(j + \lambda)
$$

$$
\le \lambda^{(\alpha-1)(s+1)/2} (C_1 \lambda^{-1} + \lambda \sum_{j=s+1}^{J_1} \beta_j j)
$$

$$
\le \lambda^{(\alpha-1)(s+1)/2} (C_1 \lambda^{-1} + \lambda \sum_{j=s+1}^{J_1} \beta_j j^2)
$$

$$
\le \lambda^{(\alpha-1)(s+1)/2} C_2
$$

$$
\le C_2 \lambda^{(3-s)/2} .
\tag{68}
$$

Let $J_2 = \lambda$. We have

$$
\sum_{j=J_1+1}^{J_2} \beta_j j(j + \lambda) v_{j-1,\lambda+1} \le \lambda^{\frac{(\alpha-1)\lambda^\alpha}{2}} C_3
$$

$$
\le C_3 \lambda^{(3-s)/2} .
\tag{69}
$$

Let $J_3 = \lambda^{3/2}$. We have

$$
\sum_{j=J_2+1}^{J_3} \beta_j j(j + \lambda) v_{j-1,\lambda+1} \le e^{-\frac{1}{2}\sqrt{\lambda}} C_4
$$

$$
\le C_4 \lambda^{(3-s)/2} .
\tag{70}
$$

Finally, the remainder satisfies

$$\sum_{j>J_3} \beta_j j(j+\lambda) v_{j-1,\lambda+1} \le C_5 (e/2)^{-\lambda}$$

$$\le C_5 \lambda^{(3-s)/2} , \tag{71}$$

which proves (67).

∎

*Proof of Lemma D.7.* We prove this result by analysing different regimes for $j$ and $\lambda$. Concretely, we claim the following:

**Claim D.9.** *We have the following regimes:*

1. *For $j = \Omega(1)$, we have*

$$v_{j,\lambda} \lesssim \left[1 - \left(\frac{\lambda}{j+\lambda}\right)^2\right]^{j/2} . \tag{72}$$

2. *For $j = \Theta(\lambda^\alpha)$, with $0 < \alpha < 1$, we have*

$$v_{j,\lambda} \lesssim \lambda^{\frac{(\alpha-1)\lambda^\alpha}{2}} . \tag{73}$$

3. *For $j = \Theta(\lambda^\alpha)$, with $1 \le \alpha < 2$, we have*

$$v_{j,\lambda} \lesssim e^{-\frac{1}{2}\lambda^{2-\alpha}} . \tag{74}$$

4. *For $j = \Omega(\lambda^\alpha)$, with $\alpha > 3/2$, we have*

$$v_{j,\lambda} \lesssim e^{-\lambda} . \tag{75}$$

To prove the first three regimes of Claim D.9, we control $v_{j,\lambda}$ based on the distribution of the roots of $P_{j,\lambda}$. We recall that $(z_{k,j,\lambda})_{k\le j}$ denotes the roots of $P_{j,\lambda}$ in increasing order, and $z_{j,\lambda} = z_{j,j,\lambda}$ its largest root.

**Lemma D.10** (Representation of $P_{j,\lambda}$ in terms of its roots, [De Carli, 2008, Lemma 2.1]). *We have*

$$P_{j,\lambda}(t) = \begin{cases} \prod_{k=j/2}^{j} \frac{t^2 - z_{k,j,\lambda}^2}{1 - z_{k,j,\lambda}^2} & \text{if } j \text{ even,} \\ t \prod_{k=(j+1)/2}^{j} \frac{t^2 - z_{k,j,\lambda}^2}{1 - z_{k,j,\lambda}^2} & \text{if } j \text{ odd .} \end{cases} \tag{76}$$

From this representation, we deduce that $v_{j,\lambda}$ can be calculated explicitly. Indeed, as the local maxima of $|P_{j,\lambda}(t)|$ are increasing Szego [1939], [DLMF, Eq (18.14.15)], we have the following equation:

$$v_{j,\lambda} = -P_{j,\lambda}(z_{j-1,\lambda+1}) = - \begin{cases} \prod_{k=j/2}^{j} \frac{z_{j-1,\lambda+1}^2 - z_{k,j,\lambda}^2}{1 - z_{k,j,\lambda}^2} & \text{if } j \text{ even,} \\ z_{j-1,\lambda+1} \prod_{k=(j+1)/2}^{j} \frac{z_{j-1,\lambda+1}^2 - z_{k,j,\lambda}^2}{1 - z_{k,j,\lambda}^2} & \text{if } j \text{ odd .} \end{cases} \tag{77}$$

Let us focus first on the case $j$ even, for simplicity. We can rewrite (77) more conveniently as

$$v_{j,\lambda} = \frac{z_{j,\lambda}^2 - z_{j-1,\lambda+1}^2}{1 - z_{j,\lambda}^2} \prod_{k=j/2}^{j-1} \frac{z_{j-1,\lambda+1}^2 - z_{k,j,\lambda}^2}{1 - z_{k,j,\lambda}^2} .$$

For $\delta \in (0, z_{j-1,\lambda+1})$ let

$$m(\delta, j, \lambda) := |\{k \in \{j/2, j\}; z_{k,j,\lambda} \ge \delta\}|$$

denote the number of zeros of $P_{j,\lambda}$ in the interval $(\delta, 1)$. Since the function $t \mapsto \frac{a^2 - t^2}{1 - t^2}$ is decreasing in $t \in (0, a)$, we have

**Fact D.11.** *We have the upper bound:*

$$v_{j,\lambda} \leq \frac{z_{j,\lambda}^2 - z_{j-1,\lambda+1}^2}{1 - z_{j,\lambda}^2} \inf_{\delta} \left( \frac{z_{j-1,\lambda+1}^2 - \delta^2}{1 - \delta^2} \right)^{m(\delta,j,\lambda)} . \tag{78}$$

Letting $\delta = z_{j/2,j,\lambda}$ the smallest positive root of $P_{j,\lambda}$ we have

$$v_{j,\lambda} \leq \frac{z_{j,\lambda}^2 - z_{j-1,\lambda+1}^2}{1 - z_{j,\lambda}^2} \left( \frac{z_{j-1,\lambda+1}^2 - z_{j/2,j,\lambda}^2}{1 - z_{j/2,j,\lambda}^2} \right)^{j/2} . \tag{79}$$

We can thus obtain an explicit control on $v_{j,\lambda}$ from bounds on the zeros of the Gegenbauer polynomials. We complement the upper bound on the largest root (Fact D.4) with lower bounds for all positive roots, as well as a sharp lower bound for its largest root Dimitrov and Nikolov [2010]:

**Theorem D.12** (Upper and Lower bounds for Gegenbauer roots, [Dimitrov and Nikolov, 2010, Theorem 2]). *Let*
$$b_{j,\lambda} = j^3 + 2(\lambda - 1)j^2 - (3\lambda - 5)j + 4(\lambda - 1) ,$$
$$a_{j,\lambda} = 2(j + \lambda - 1)(j^2 + j(\lambda - 1) + 4(\lambda + 1)) \text{ and}$$
$$c_{j,\lambda} = j^2(j + 2\lambda)^2 + (2\lambda + 1)(j^2 + 2(\lambda + 3)j + 8(\lambda - 1)) .$$
*Then for every $k, j, \lambda$ we have*
$$\frac{b_{j,\lambda} - (j - 2)\sqrt{c_{j,\lambda}}}{a_{j,\lambda}} \leq z_{k,j,\lambda}^2 \leq \frac{b_{j,\lambda} + (j - 2)\sqrt{c_{j,\lambda}}}{a_{j,\lambda}} . \tag{80}$$

**Theorem D.13** (Lower bound for largest root, [Driver and Jordaan, 2012, Section 2.3]).
$$z_{j,\lambda}^2 > 1 - \frac{(2\lambda + 1)(2\lambda + 3)}{(j - 1)(j + 2\lambda + 1) + (2\lambda + 1)(2\lambda + 3)} := 1 - \frac{g_{j,\lambda}}{h_{j,\lambda}} . \tag{81}$$

Rewriting Fact D.4 as $z_{j,\lambda}^2 \leq \frac{e_{j,\lambda}}{f_{j,\lambda}}$ , with
$$e_{j,\lambda} = (j - 1)(j + 2\lambda - 2) , \quad f_{j,\lambda} = (j + \lambda - 2)(j + \lambda - 1) ,$$
and using again the monotonocity of $t \mapsto \frac{t-p}{1-t}$ we can bound the first term in the RHS of (79) as
$$\frac{z_{j,\lambda}^2 - z_{j-1,\lambda+1}^2}{1 - z_{j,\lambda}^2} \leq \frac{e_{j,\lambda}/f_{j,\lambda} + g_{j-1,\lambda+1}/h_{j-1,\lambda+1} - 1}{1 - e_{j,\lambda}/f_{j,\lambda}} . \tag{82}$$

For $j, \lambda = \omega(1)$, we have
$$a_{j,\lambda} \simeq 2j(j + \lambda)^2, \qquad b_{j,\lambda} \simeq j^2(j + 2\lambda) , \quad \sqrt{c_{j,\lambda}} \simeq j(j + 2\lambda) ,$$
$$e_{j,\lambda} \simeq j(j + 2\lambda), \qquad f_{j,\lambda} \simeq (j + \lambda)^2 ,$$
$$g_{j,\lambda} \simeq 4\lambda^2, \qquad h_{j,\lambda} \simeq j(j + 2\lambda) + 4\lambda^2 ,$$
and thus
$$\frac{z_{j,\lambda}^2 - z_{j-1,\lambda+1}^2}{1 - z_{j,\lambda}^2} \lesssim \frac{3j(j + 2\lambda)}{j(j + 2\lambda) + 4\lambda^2} \leq 3 . \tag{83}$$

Therefore,
$$v_{j,\lambda} \leq 3 \left( \frac{\frac{e_{j-1,\lambda+1}}{f_{j-1,\lambda+1}} - \frac{b_{j,\lambda} - (j-2)\sqrt{c_{j,\lambda}}}{a_{j,\lambda}}}{1 - \frac{b_{j,\lambda} - (j-2)\sqrt{c_{j,\lambda}}}{a_{j,\lambda}}} \right)^{j/2}$$

$$\leq 3 \left( \frac{a_{j,\lambda}e_{j-1,\lambda+1} - f_{j-1,\lambda+1}(b_{j,\lambda} - (j - 2)\sqrt{c_{j,\lambda}})}{f_{j-1,\lambda+1}(a_{j,\lambda} - b_{j,\lambda} + (j - 2)\sqrt{c_{j,\lambda}})} \right)^{j/2}$$

$$= 3 \left( \frac{2j(j + \lambda)^2 j(j + 2\lambda) - (j + \lambda)^2(j^2(j + 2\lambda) - j^2(j + 2\lambda))}{(j + \lambda)^2(2j(j + \lambda)^2 - j^2(j + 2\lambda) + j^2(j + 2\lambda))} \cdot (1 + o_{j,\lambda}(1)) \right)^{j/2}$$

$$\lesssim \left( \frac{j(j + 2\lambda)}{(j + \lambda)^2} \right)^{j/2}$$

$$= \left[ 1 - \left( \frac{\lambda}{j + \lambda} \right)^2 \right]^{j/2} . \tag{84}$$

As a direct consequence of (84), we immediately obtain Eqs (72), (73) and (74). The case where $j$ is odd is treated analogously.

Let us now study the regime $j = \omega(\lambda^{3/2})$. Given $z \in \mathbb{C}$ with $|z| < 1$, Gegenbauer polynomials admit the following generating function [Watson, 1922, Section 3.32]:

$$\frac{1}{(1 - 2z\cos\theta + z^2)^\lambda} = \sum_{j \geq 0} \bar{P}_{j,\lambda}(\cos\theta)z^j . \tag{85}$$

From this generating function, the Cauchy integral formula leads to the following integral representation:

**Fact D.14** ([Ursell, 2007, Eq (1.2)]). *For any $0 < \rho < 1$, we have*

$$\bar{P}_{j,\lambda}(\cos\theta) = \frac{1}{2\pi i} \oint_{|z|=\rho} \frac{dz}{(1 - 2z\cos\theta + z^2)^\lambda z^{j+1}} . \tag{86}$$

Assume $j = \Theta(\lambda^\alpha)$, with $\alpha > 3/2$. We are interested in the above representation for $\bar{\theta} = \arccos(z_{j-1,\lambda+1})$. From Theorem D.13, we have $z_{j-1,\lambda+1}^2 \geq 1 - d_{j,\lambda}/(2c_{j,\lambda})$, and thus

$$\bar{\theta}^2 \lesssim \frac{d_{j,\lambda}}{2c_{j,\lambda}} \simeq \frac{32\lambda^2 j^4}{16j^6} = \frac{2\lambda^2}{j^2} ,$$

so $\bar{\theta} = O(\lambda/j)$. Combining this upper bound with the lower bound obtained from Fact D.4 we have $\bar{\theta} = \Theta(\lambda/j)$.

Using $1 - \cos\theta \simeq \theta^2/2 \simeq \lambda^2/j^2$ and

$$\begin{aligned}
|1 - 2z\cos\theta + z^2| &= \left|(1-z)^2 + 2z(1-\cos\theta)\right| \\
&\geq |1-z|^2 - 2|z|(1-\cos\theta) \\
&\geq 1 - \rho\left(2 + \Theta\left(\frac{\lambda^2}{j^2}\right)\right) + \rho^2 ,
\end{aligned} \tag{87}$$

we have

$$|\bar{P}_{j,\lambda}(\cos\theta)| \leq \inf_{0<\rho<1} |\rho|^{-(j+1)}\left(1 - \rho(2 + c\lambda^2/j^2) + \rho^2\right)^{-\lambda} := g(\rho) . \tag{88}$$

Optimizing the RHS over $\rho$ we obtain $\rho^* = \frac{j - (\sqrt{2}-1)\lambda}{j+2\lambda}$; substituting, we obtain

$$g(\rho^*) \simeq e^{-(1+\sqrt{2})\lambda}\left(\frac{j+2\lambda}{\lambda(1+\sqrt{2})}\right)^{2\lambda} . \tag{89}$$

As a result, it follows that

$$P_{j,\lambda}(\cos\theta) = \bar{P}_{j,\lambda}(\cos\theta)\frac{j!(2\lambda-1)!}{(2\lambda+j-1)!} \tag{90}$$

satisfies, for $\theta = \Theta(\lambda/j)$ and $j = \omega(\lambda^{3/2})$,

$$\begin{aligned}
\log|P_{j,\lambda}(\cos\theta)| \simeq &\, j\log j - j + 2\lambda\log(2\lambda) - 2\lambda - (j+2\lambda)\log(2\lambda+j) + 2\lambda + j \\
&- (1+\sqrt{2})\lambda + 2\lambda\log(j+2\lambda) - 2\lambda\log(\lambda(1+\sqrt{2})) \\
\simeq &- (1+\sqrt{2})\lambda ,
\end{aligned} \tag{91}$$

where we have used Stirling's approximation. This proves Eq (75) and completes the proof of Lemma D.7. ∎

*Proof of Lemma D.8.* We have, using Fact D.3, that

$$\begin{aligned}
\mathbb{E}_\rho[\mathbb{E}_{u_d}(\phi^{(r)'})^2] &= \frac{\Omega_{d-1}}{(d-1)^2\Omega_{d-2}}\sum_j \mathbb{E}_\rho\left[\bar{\alpha}_{j,d,r}^2 (j(j+d-2))^2\right] \\
&\geq \frac{\Omega_{d-1}}{\Omega_{d-2}}\sum_j j^2 \mathbb{E}_\rho\left[\bar{\alpha}_{j,d,r}^2\right] .
\end{aligned} \tag{92}$$

And we can upper bound via

$$\mathbb{E}_\rho[\mathbb{E}_{u_d}(\phi^{(r)'})^2] = \mathbb{E}_\rho[r^2\mathbb{E}_{u_d}((\phi')^{(r)})^2]$$
$$= \mathbb{E}_\rho\mathbb{E}_{x_1|\|x\|=r}[r^2(\phi(x_1)')^2]$$
$$\leq \sqrt{\mathbb{E}_\rho[r^4]\mathbb{E}_\eta(\phi')^4} \ , \tag{93}$$

where this last line is finite by our assumptions on $\phi$ and $\rho$,

so from (92) we conclude that

$$\sum_j j^2\beta_{j,d} \leq \frac{\Omega_{d-2}}{\Omega_{d-1}}\sqrt{\mathbb{E}_\rho[r^4]\mathbb{E}_\eta(\phi')^4} \ . \tag{94}$$

∎

# E  The LPG property in the non-symmetric case: proofs of Section 4.2

## E.1  Proof of Proposition 4.11

**Assumption 4.9** (Regularity of link function). *We assume that $\phi, \phi'$ are both $B$-Lipschitz, and that $\phi''(t) = O(1/t)$.*

**Assumption 4.10** (Subgaussianity). *The data distribution $\nu$ is $M$-subgaussian: for any $v \in \mathcal{S}_{d-1}$, we have $\|x \cdot v\|_{\psi_2} \leq M$, where $\|z\|_{\psi_2} := \inf\{t > 0; \mathbb{E}[\exp(z^2/t^2) \leq 2\}$ is the Orlitz-2 norm.*

**Proposition 4.11** (Uniform gradient approximation). *Under Assumptions 4.9 and 4.10, for all $\theta \in \mathcal{S}_{d-1}$,*

$$\Delta_{\nabla L}(\theta) = (1 - m^2)O\left(\widetilde{W}_{1,2}(\nu, \gamma)\log(\widetilde{W}_{1,2}(\nu, \gamma)^{-1})\right) \tag{22}$$

*where the $O(\cdot)$ notation only hides constants appearing in Assumptions 4.9 and 4.10.*

*Proof.* Recall the notation $\phi_\theta(x) = \phi(\langle x, \theta\rangle)$. Let $v = \theta^* - m\theta$. From the definition, we have that

$$\langle\nabla_\theta^\mathbb{S}L(\theta), \theta^*\rangle = 2\mathbb{E}_\nu\left[\phi_\theta'(\phi_\theta - \phi_{\theta^*})(x \cdot v)\right]$$
$$:= \mathbb{E}_\nu[g_{\theta,\theta^*}] \ . \tag{95}$$

Since $\mathbb{E}_\gamma[g_{\theta,\theta^*}]$ is precisely $\bar\ell'(m)(1 - m^2)$, we need to establish that

$$\sup_\theta |\mathbb{E}_\nu g_{\theta,\theta^*} - \mathbb{E}_\gamma g_{\theta,\theta^*}| \leq C\sqrt{1 - m^2}\widetilde{W}_{1,2}(\nu, \gamma)\left(\log\widetilde{W}_{1,2}(\nu, \gamma)\right)^2 \ . \tag{96}$$

Fix $\theta$ and let $P_{\theta,\theta^*}$ be the orthogonal projection onto the subspace spanned by $\theta, \theta^*$. For $R > 0$ we consider $A_R = \{x \in \mathbb{R}^d; \|P_{\theta,\theta^*}x\| \leq R\}$.

$$|\mathbb{E}_\nu g_{\theta,\theta^*} - \mathbb{E}_\gamma g_{\theta,\theta^*}| = \left|\int g_{\theta,\theta^*}(x)(\nu(dx) - \gamma(dx))\right|$$
$$\leq \underbrace{\left|\int_{x \in A_R} g_{\theta,\theta^*}(x)(\nu(dx) - \gamma(dx))\right|}_{T_a} + \underbrace{\left|\int_{x \notin A_R} g_{\theta,\theta^*}(x)(\nu(dx) - \gamma(dx))\right|}_{T_b} \ . \tag{97}$$

Let us first bound $T_a$. Denote by $v = \theta^* - m\theta$, with $\|v\|^2 = 1 - m^2$ Since $\phi$ and $\phi'$ are Lipschitz and $|\phi''| \leq O((1 + t)^{-1})$ by Assumption 4.9, we have that

$$\nabla_x g_{\theta,\theta^*}(x) = \phi_\theta''(\phi_\theta - \phi_{\theta^*})x^\top v\theta + \phi_\theta'(\phi_\theta'\theta - \phi_{\theta^*}'\theta^*)x^\top v + \phi_\theta'(\phi_\theta - \phi_{\theta^*})v \tag{98}$$

satisfies

$$\|\nabla_x g_{\theta,\theta^*}(x)\| \leq 2\|v\|C\mathrm{Lip}(\phi)R + 4\|v\|\mathrm{Lip}(\phi)^2R$$
$$\leq C\|v\|R \ , \tag{99}$$

and as a result we have that $g_{\theta,\theta^*}$ is $C\|v\|R$-Lipschitz when restricted to $A_R$, and thus

$$T_a \leq CR\|v\|\widetilde{W}_{1,2}(\nu,\mu) \ . \tag{100}$$

Let us now control the tail $T_b$. Since $x^\top v$ is $\sqrt{2}M\|v\|$-subgaussian and $\phi$ is Lipschitz, we have that $z = |g_{\theta,\theta^*}(x)|$ is $\tilde{M}\|v\|$-subexponential where $\tilde{M}$ only depends on $M$ and $L$. It follows that

$$\begin{aligned}
T_b &\leq R(\mathbb{P}_\nu(z \geq R) + \mathbb{P}_\gamma(z \geq R)) \\
&\leq R\exp\left(-\frac{\beta}{\|v\|}R\right) \ ,
\end{aligned} \tag{101}$$

where $\beta$ is a constant that depends only on $\tilde{M}$. As a result, we have

$$|\mathbb{E}_\nu g_{\theta,\theta^*} - \mathbb{E}_\gamma g_{\theta,\theta*}| \leq \inf_{R>0}\left(CR\|v\|\widetilde{W}_{1,2}(\nu,\mu) + R\exp\left(-\frac{\beta}{\|v\|}R\right)\right) \ . \tag{102}$$

Setting

$$R = -\|v\|\beta^{-1}\log((C\|v\|\widetilde{W}_{1,2}(\nu,\gamma)))$$

we obtain

$$\begin{aligned}
|\mathbb{E}_\nu g_{\theta,\theta^*} - \mathbb{E}_\gamma g_{\theta,\theta*}| &\leq 2C(1-m^2)\beta^{-1}\left|\log(C\|v\|\widetilde{W}_{1,2}(\nu,\gamma))\right|\widetilde{W}_{1,2}(\nu,\gamma) \\
&\leq (1-m^2)O\left(\widetilde{W}_{1,2}(\nu,\gamma)\log(\widetilde{W}_{1,2}(\nu,\gamma)^{-1})\right) \ ,
\end{aligned} \tag{103}$$

as claimed.

∎

## E.2  Proof of Proposition 4.14

We leverage Proposition 4.11 and the fact that if $\phi$ has information exponent $s = 2$, then $\bar{\ell}'(m) \simeq m$ for small $m$.

We need to show that for $b = \Theta(\log d)$ we have

$$\langle\nabla_\theta L(\theta), \theta^*\rangle \geq C\left(m - \frac{b}{\sqrt{d}}\right) \ , \ \text{for} \ \frac{b}{\sqrt{d}} \leq m \leq \frac{1}{2} \ , \tag{104}$$

as well as

$$\langle\nabla_\theta L(\theta), \theta^*\rangle \geq C'(1-m^2) \tag{105}$$

for $m \geq \frac{1}{2}$.

From (103) and $\widetilde{W}_{1,2}(\nu,\gamma) \leq C/\sqrt{d}$, we obtain

$$\begin{aligned}
\langle\nabla_\theta L(\theta), \theta^*\rangle &= \bar{\ell}'(m)(1-m^2) + \langle\nabla_\theta L(\theta), \theta^*\rangle - \bar{\ell}'(m)(1-m^2) \\
&\geq 2\alpha_2^2 m(1-m^2) - (1-m^2)\tilde{C}\widetilde{W}_{1,2}(\nu,\gamma)\log(\widetilde{W}_{1,2}(\nu,\gamma)^{-1}) \\
&\geq \left(\alpha_2^2 m - \tilde{C}\frac{C}{\sqrt{d}}\log(\sqrt{d}/C)\right)(1-m^2) \\
&\geq \alpha_2^2\left(m - \tilde{C}\frac{C}{\alpha_2^2\sqrt{d}}\log(\sqrt{d}/C)\right)(1-m^2) \\
&\geq \alpha_2^2\left(m - \frac{\log d^{C'}}{\sqrt{d}}\right)(1-m^2) \ ,
\end{aligned} \tag{106}$$

which proves (104) and (105).

## E.3  Proof of Proposition 4.16

**Assumption 4.15** (Additional Regularity in third derivatives). $\phi$ *admits four derivatives bounded by $L$, with $|\phi^{(3)}(t)| = O(1/t)$ and $|\phi^{(4)}(t)| = O(1/t^2)$. Moreover, the third moment of the data distribution is finite: $\tau_3 = \mathbb{E}_{t\sim\eta}[t^3] < \infty$.*

**Proposition 4.16** (Stein's method for product measure). *Let $\chi(\theta, \theta^*) := \|\theta\|_4^2 + \|\theta^*\|_4^2$. Under Assumptions 4.9, 4.10 and 4.15, there exists a universal constant $C = C(M, B, \tau_3)$ such that*

$$\Delta_L(\theta) \leq C\chi(\theta, \theta^*)\,,\text{ and }\,\Delta_{\nabla L}(\theta) \leq C\sqrt{1 - m^2}\chi(\theta, \theta^*)\,. \tag{23}$$

*Proof.* Recall the notation $\phi_\theta(x) = \phi(\langle x, \theta\rangle)$, and, using $v = \theta^* - m\theta$,

$$h_{\theta,\theta^*}(x) := \phi_\theta^2 - 2\phi_\theta\phi_{\theta^*}\,, \tag{107}$$

$$g_{\theta,\theta^*}(x) := 2\phi_\theta'(\phi_\theta - \phi_{\theta^*})(x \cdot v)\,, \tag{108}$$

so that

$$\Delta_L(\theta) = \mathbb{E}_\nu[h_{\theta,\theta^*}(x)] - \mathbb{E}_\gamma[h_{\theta,\theta^*}(x)]\,, \tag{109}$$

$$\Delta_{\nabla L}(\theta) = \mathbb{E}_\nu[g_{\theta,\theta^*}(x)] - \mathbb{E}_\gamma[g_{\theta,\theta^*}(x)]\,. \tag{110}$$

The result is obtained via the following Stein coupling method for product measures:

**Theorem E.1** (Stein Coupling, [Röllin, 2013, Theorem 3.1]). *Let $X$ be a $d$-dimensional random vector of independent coordinates, such that $\mathbb{E}X = 0$, $\mathbb{E}[XX^\top] = I_d$ and $\mathbb{E}|X_i|^3 = \tau_i^3 < \infty$. If $Z$ is a standard Gaussian random vector, and $h : \mathbb{R}^d \to \mathbb{R}$ is three-times differentiable, then*

$$|\mathbb{E}h(X) - \mathbb{E}h(Z)| \leq \frac{5}{6}\sum_{i=1}^d \tau_i^3\|\partial_{x_i}^3 h\|_\infty\,. \tag{111}$$

We verify that, thanks to the decay assumptions in Assumption 4.15, we have

$$\partial_{x_i}^3 g_{\theta,\theta^*}(x) = \lambda_1(x)\theta_i^3 + \lambda_2(x)\theta_i^2\theta_i^* + \lambda_3(x)\theta_i(\theta_i^*)^2 + \lambda_4(x)(\theta_i^*)^3\,, \tag{112}$$

$$\partial_{x_i}^3 h_{\theta,\theta^*}(x) = \lambda_5(x)\theta_i^3 + \lambda_6(x)\theta_i^2\theta_i^* + \lambda_7(x)\theta_i(\theta_i^*)^2 + \lambda_8(x)(\theta_i^*)^3\,, \tag{113}$$

where

$$\sup_{k\in\{1,2,3,4\}}|\lambda_k(x)| \leq C\|v\|\,,\quad \sup_{k\in\{5,6,7,8\}}|\lambda_k(x)| \leq \tilde{C}\,. \tag{114}$$

Observing by Cauchy-Schwartz that

$$\max\left\{\sum_i |\theta_i|^2|\theta_i^*|, \sum_i |\theta_i|^3\right\} \leq \|\theta\|_4^2\,,$$

$$\max\left\{\sum_i |\theta_i^*|^2|\theta_i|, \sum_i |\theta_i^*|^3\right\} \leq \|\theta^*\|_4^2\,,$$

we obtain from Theorem E.1 that

$$\Delta_L(\theta) \leq C\chi(\theta, \theta^*)\,,\ \Delta_{\nabla L}(\theta) \leq C'\|v\|\chi(\theta, \theta^*)\,,$$

as claimed. ∎