# OpenReview forum: "On Single-Index Models beyond Gaussian Data"
_NeurIPS.cc/2023/Conference — NeurIPS 2023 poster_

### Official Review · Reviewer_CJfX · 2023-06-22

**Soundness:** 3 good
**Presentation:** 4 excellent
**Contribution:** 2 fair
**Rating:** 6
**Confidence:** 4

**Summary:**

This paper studies the problem of learning a single-index model beyond Gaussian data. Prior work (Ben Arous et al. 2021) studies the optimization landscape of recovering the hidden direction $\theta^*$ given a known link function $\phi$ when the covariates are drawn from the standard Gaussian distribution; in this case, the sample complexity is governed by the information exponent of $\phi$. This paper’s key contribution is defining the “Local Polynomial Growth” condition, which characterizes the geometry of the population loss landscape. The main results, Theorems 3 and 4, show that online SGD recovers the hidden direction with sample complexity depending on the order parameter in the LPG condition. The proof of this main theorem relies on the same martingale-plus-drift decomposition from (Ben Arous et al. 2021). The paper concludes with specific examples of data distributions satisfying the LPG condition; for example, spherically symmetric distributions.

**Strengths:**

- The paper is well-written and easy to understand.
- Learning a single-index model is a well studied problem in the statistics literature and is a primitive for neural network learning. Prior work in this area either assumes Gaussian data or that the link function satisfies basic properties such as monotonicity. This paper initiates the study of the landscape properties of learning single-index models with arbitrary data distribution and link function, which is to the best of my knowledge novel and an important direction of future work.
- The local polynomial growth definition is an interesting and useful way to abstract away the key ideas of the proof in (Ben Arous et al. 2021), and has not appeared previously in the literature. This definition allows for their key idea of escaping a higher-order saddle at initialization to apply to a larger family of distributions more generally.
- Section 4.1 shows how the LPG framework can be used to learn single index models over arbitrary spherically symmetric distributions, which is a novel and interesting result.

**Weaknesses:**

- There is one major weakness I am concerned about. Theorem 3 depends on this constant $C$, which in turn depends on a constant $K$ defined in Appendix B. This constant is used to upper bound quantities such as $\sup_\theta E_x\|\nabla M(x, \theta)\|^2$ which show up when concentrating the martingale term. However, it appears to me that this constant is not actually dimension independent. In the proof of Lemma 22, before line 558, this constant depends on $|x|^2$ which is typically $\Theta(d)$ for many examples (such as Gaussian or uniform on sphere of radius $\sqrt{d}$). Thus it seems that there is an additional dimension dependence hidden in the constant $C$. Can you please clarify whether there is indeed a hidden dimension dependence here? This is a key issue because it changes the exponent of $d$ in Theorem 3. In my opinion this is a major issue that makes me recommend reject, but if you can clarify this point or I have misunderstood I am wiling to increase my score.
- A more minor weakness is that I find the examples in Section 4.2 to be a bit restrictive. For instance, requiring *all* 2d marginals of $\nu$ to be close to Gaussian in Wasserstein distance seems to be a very restrictive assumption, and I am not sure what distributions satisfy this property (beyond uniform on the sphere of radius $\sqrt{d}$). Next, proposition 15 shows that LPG is satisfied only for $b = \log d/\sqrt{d}$, which seems to imply that polynomially many initializations would need to be sampled before a good one is reached. Finally, as the authors indeed point out, the Stein’s method argument only shows the LPG along dense directions and does not yield an end-to-end guarantee for online SGD.

**Questions:**

- As mentioned above, can you please comment on the dimension dependence of the constant $C$ in Theorem 3?
- A key requirement in the paper is for the initialization to have sufficient correlation with $\theta^*$. How does one check whether an initialization has sufficient correlation? This is particularly of interest for the 2D-Wasserstein distance example, where the probability of achieving a good initialization is polynomially small.
- Do you have examples of specific distributions which satisfy Assumption 14?
- lines 283-284: “by paying an additional $O(d^2)$ cost in complexity” — what do you mean by this? is this just the complexity of resampling until a good initialization is achieved?

**Limitations:**

Limitations/broader impact are adequately addressed.

---

> ### Author Rebuttal · Authors · 2023-08-09
>
> We thank very much the reviewer for the sharp comments and discussion that will help improve the paper greatly.
>
> - *Dimension dependent constants.* This is the major concern of the reviewer, and we would like to thank him for the important comment: indeed constants like $\sup_{\theta} E[\|\nabla M(x, \theta)\|^2]$ typically depend on the dimension as $O(d)$ (e.g. gaussian data) and the way the proof was written in the Neurips submission is wrong! However, we want to emphasise that this is only a technical flaw that can be easily circumvented by the following: in fact, the correct object that we need to upper bound carefully is the variance of *directional martingale* $\sup_\theta E[\langle \nabla M(x, \theta), \theta_*\rangle^2]$ (and not its norm) which, this time, does not depend on the dimension (e.g. for approximately gaussian or spherically symmetric data)! This corresponds to equation 1.3 of Ben Arous et al. paper, and we were too loose on this argument by bounding directly quantity like $\langle \nabla M(x, \theta), \theta_*\rangle$ with $\| \nabla M(x, \theta)\|$. We have fixed this in the proof in the Appendix and clearly acknowledge the previous flaw.
>
> - *Section 4.2.* It is true that this section, which makes a perturbative statement on the distribution, suffers from the limitations that the reviewer mention. However, we see this section as paving the way of the description that offers Stein's Lemma for product measures. The aim of this section is hence to provide the basis of a promising route in order to understand the universality of the Gaussian for this problem.
>
> - *Initialization.* For the sake of clarity, we repeat the comment made for another reviewer. For line 161, we wanted to say that if we initialized randomly a batch of different $\theta_0$, then the probability of one being well initialized (i.e. success) will be lowered significantly.  As it is *a priori* impossible to measure the alignment without the knowledge of $\theta_*$, what can be done in practice is to run in parallel all the dynamics with different initializations and to take, at the end of training, the one with smallest empirical error (wrt the samples used for SGD). We will clarify this.
>
> - *Distribution satisfying Assumption 14.* There is a discussion on such distributions given from line 294 to line 303. Furthermore, as already explained, even if this is not the case uniformly, for product measures,  Proposition 17 also tells us that typically, a LPG property is also expected.

---

> > ### Comment · Reviewer_CJfX · 2023-08-11
> > **Response to the Authors**
> >
> > Thank you for your response, and for helping to clarifying my concern re the dimension dependence. So if I understand correctly, your claim is that this issue can be resolved by making the assumption that $\sup_\theta E[(\nabla M(x, \theta) \cdot \theta^*)^2]$ is upper bounded by a constant (akin to Assumption B in Ben Arous. et al)? I see how this lets you bound some of the terms, such as for Lemma 20, but it is not clear to me whether it is sufficient for some of the other terms. For example, the quantity $\xi(x, \theta)$ scales with $|\nabla M(x, \theta)|^2$ and so its not clear that $\sup_\theta E_x[\xi(x, \theta)]$ can be bounded by a constant. If you can provide a sketch of the argument for why this assumption allows you to circumvent such technical issues that would be greatly appreciated and give me more confidence in the correctness of your proof.

---

> > > ### Author Response · Authors · 2023-08-11
> > >
> > > Thanks for keeping on the discussion. You are totally right: $\mathrm{sup}_\theta \mathbb{E}_x[ \xi (x, \theta)] $ blows up with $d$ as well, and this time, it is inevitable. Since your first comment, we have hence worked to totally solved this.
> > >
> > > The overall argument is that the term corresponding to $\xi(x, \theta)$ comes from the discretization and has a $\delta^2$ dependency (compared to $\delta$), so that the additional $\delta$ factor can "absorb" the additional $d$ term. This is the main idea why the proof works, however since it seems that the reviewer wants the precise and deep reason (and we thank him for this!)... technically it is a bit more complicated.
> > >
> > > Let us detail this. In fact,  $\xi$ can be divided into two terms one that will have a $\delta^2$ and one with a $\delta^3$, i.e. to fix notations $ \delta^2 \xi(x, \theta) = \delta^2 \xi_1 (x, \theta) + \delta^3 \xi_2 (x, \theta) $.
> > >
> > > - For the second part, i.e. $\delta^3  \xi_2 (x, \theta) $, the basic argument depicted earlier holds. As in any case $\delta < 1/d$ simple Markov inequality provides the bound we want even-though $\mathbb{E}_x( \xi_2 (x, \theta) ) = O(d^{1+a})$, with some small $a > 0$.
> > > - For the first part, things are bit more complicated: it is necessary once again to divide it as a martingale increment $\delta^2(\xi_1 (x, \theta) - \mathbb{E} [\xi_1 (x, \theta)])$ and a drift $\delta^2 \mathbb{E} [\xi_1 (x, \theta)] \sim \delta^2 d \ll 1/(10\sqrt{d}) $. This drift has the correct scale. For the martingale term, applying Doob's maximal inequality gives that the martingale term is smaller than $1/(10\sqrt{d}) $ with probability at least $1 - T \delta^4 d \mathbb{E} [\xi_1 (x, \theta)^2 ] \sim T \delta^4 d^3 \leq T \delta^2 d $ , which is the scale we tolerate and we have already for the true directional martingale term!
> > >
> > > In conclusion, we thought to give the precise arguments of the rewritten proof in response to the acute comments of the reviewer. Please also remark that even if the proof is slightly different that the proof of Ben Arous et al, 2021, morally speaking, the result has to go through as it went for this paper.  We are happy to discuss any detail with the reviewer during the discussion period.

---

> > > > ### Comment · Reviewer_CJfX · 2023-08-17
> > > > **Response**
> > > >
> > > > Thank you very much to the authors for their response. The proof sketch you have given mostly makes sense to me. There is one point though I am still confused by. You state that the $\xi_1$ drift term scales like $\delta^2\mathbb{E}[\xi_1(x, \theta)] \sim \delta^2d$. But since we need to sum this term over $T$ timesteps (the last term in equation 26), won't the size of this term become $\delta^2dT$ which is now $\Theta(1)$?

---

> > > > > ### Author Response · Authors · 2023-08-18
> > > > > **Precisions on the technical argument.**
> > > > >
> > > > > Absolutely, this was simply a sketch, and we were trying to give the overall argument. The reviewer is right that there is a last trick that needs to be performed to ensure the correct bound. At this point, we hope that the "only" term that needs a clarification is the $\delta^2 \xi_1$ term mentioned above.
> > > > >
> > > > > First, notice that we will use its precise shape, and as the reviewer will see: it is important that it scales with $m_t$. This is the reason why the previous argument heuristically holds ($m_t \sim 1/\sqrt{d}$ in the first phase gives the correct order of magnitude).
> > > > >
> > > > > But let us dive more deeply into the technical reason.  First, let us explicitate this term totally: it corresponds in fact to  $ \xi_1(\theta_t, x_t) = |m_t| |\nabla l(\theta_t, x_t)|^2$. However, to apply some (Freedman) concentration on the submartingale that will result from this, we need first to threshold it and divide it as
> > > > > $$  |m_t| |\nabla l(\theta_t, x_t)|^2= |m_t| |\nabla l(\theta_t, x_t)|^2\ 1_{ |\nabla l(\theta_t, x_t)|^2 > M} + |m_t| |\nabla l(\theta_t, x_t)|^21_{|\nabla l(\theta_t, x_t)|^2 \leq M}, $$
> > > > > where we introduced some parameter $M>0$ that we will take of scale $d \sqrt{d}$.
> > > > >
> > > > > - **First term**. The first term $|m_t| |\nabla l(\theta_t, x_t)|^2\ 1_{ |\nabla l(\theta_t, x_t)|^2 > M}$ is handled via a time-union bound and simply Markov inequality ( + Cauchy-Schwartz) as
> > > > > $$ \mathbb{P}\left(\delta^2  |m_t| |\nabla l(\theta_t, x_t)|^2\ 1_{ |\nabla l(\theta_t, x_t)|^2 > M} > a/\sqrt{d}\right) \leq  \frac{\delta^2 \sqrt{d}}{a} \mathbb{E}\left[|\nabla l|^2\ 1_{ |\nabla l |^2 > M}\right],$$
> > > > > via Markov inequality. Furthermore, we have
> > > > > $$\mathbb{E}\left[|\nabla l|^2\ 1_{ |\nabla l |^2 > M}\right] \leq  \sqrt{\mathbb{E}[|\nabla l|^4]}\sqrt{\mathbb{P}( |\nabla l(\theta_t, x_t)|^2 > M )} \leq \sqrt{d^2}  \sqrt{\frac{d^2}{M^2}} = \frac{d^2}{M},$$
> > > > > as $\mathbb{E}[|\nabla l|^4] \sim d^2$. Hence, finally  the initial probability is upper bounded by $ \frac{\delta^2 d^2 \sqrt{d}}{a M} $, and via a union bound, the probability that the sum of this term until time $T>0$ is larger that $a / \sqrt{d}$ is smaller than $$ \frac{\delta^2  T d^2 \sqrt{d}}{a M} = \varepsilon O(1) ,$$
> > > > > given the scale of $\delta$ that is such that $\delta^2 T d  = \varepsilon O(1)$ and as we took the threshold $ M \sim d \sqrt{d}$. This concludes the argument for the first term.
> > > > >
> > > > > - **Second term**. This second term, $|m_t| |\nabla l(\theta_t, x_t)|^2\ 1_{ |\nabla l(\theta_t, x_t)|^2 < M}$,  is to be compared with the drift that scales (up to small translation and $O(1)$ scale) as $ \delta m_t^{s-1}$. Say that we divide this drift in two terms $ \delta m_t^{s-1}/2 + \delta m_t^{s-1}/2$. We will use one of this two terms to "absorb" the negative drift by controlling $\delta m_t^{s-1}/2 - \delta^2 |m_t| |\nabla l |^2\ 1_{ |\nabla l|^2 < M}$, and we keep the second part $\delta m_t^{s-1}/2$ to make the correlation increase as already written.
> > > > >
> > > > >  Summing this up to time $T >0$, the controlled part gives the term (we forget factor $2$ as this only corresponds to consider $\delta \leftarrow \delta / 2 $):
> > > > > $$ S_T = \delta \sum_{k = 1}^T m_k^{s-1} \left( 1 - \frac{\delta}{m_k^{s-2}} |\nabla l(\theta_k, x_k)|^2\ 1_{ |\nabla l(\theta_k, x_k)|^2 < M}
> > > > >  \right).$$
> > > > > In fact, for $m_t \gtrsim 1/\sqrt{d}$ (that we assume by conditioning), this term is a sub-martingale if $\delta d^{{\frac{s-2}{2}}} \mathbb{E}[|\nabla l|^2] <  1, $ which is always true  as $\mathbb{E}[|\nabla l|^2] \sim d$. Furthermore, we can bound almost surely the increment (this was the purpose of the thresholding!) as $|S_{T+1} - S_T| \leq \delta (1 + \delta M)$, and the conditional variance increment as $$ \mathbb{E}[ (S_{T+1} - S_T)^2 | F_T ] \leq \delta^2 (1 + \delta^2 d ) \sim \delta^2. $$
> > > > > Hence, by Freedman inequality,
> > > > >  $$\mathbb{P}(S_T \leq - 1/ \sqrt{d}) \leq \exp\left(   \frac{d^{-1}}{ T \delta^2 + \delta(1 + \delta M)  d^{-1/2} }    \right) \sim  \exp\left( - \frac{O(1)}{\varepsilon} \right), $$
> > > > > because we recall that $T d \delta^2 = \varepsilon O(1)$, also $\delta d^{1/2} = o(1)$ and finally $\delta^2 M d^{1/2} \sim \delta^2 d^2 \leq \varepsilon O(1) $, in any case.
> > > > >
> > > > > --------------------------------------------------------------------------------------------------------------------------------------------------------------------------
> > > > > --------------------------------------------------------------------------------------------------------------------------------------------------------------------------
> > > > >
> > > > > **Conclusion.** The purpose of all this is to show that the remaining drift scaling as $\delta/2 \sum_{k = 1}^T m_k^{s-1}$ controls the growth of the correlation up to small errors $O(1)/\sqrt{d}$, with high probability. As the reviewer wanted us to be precise about the technical argument, this is the way that we have corrected this (quite similarly to what is done in Ben arous et al ). We are at the disposal of the reviewer for any additional precisions.

---

> > > > > > ### Comment · Reviewer_CJfX · 2023-08-20
> > > > > > **Thank you for the detailed response**
> > > > > >
> > > > > > Thank you very much to the authors for their detailed response. The proof they have presented here looks to be correct, and to the best of my knowledge fixes the technical issue in the original submission. I thus have confidence in the overall correctness of their proof, and would like to increase my score to recommend an accept.

---

> > > > > > > ### Author Response · Authors · 2023-08-21
> > > > > > >
> > > > > > > Thank you very much for your time and the precisions of your comment.
> > > > > > > We'll thank your contribution to improve the paper in the revised version.

---

### Official Review · Reviewer_PMRM · 2023-07-04

**Soundness:** 3 good
**Presentation:** 3 good
**Contribution:** 2 fair
**Rating:** 6
**Confidence:** 4

**Summary:**

This paper extends the results of Ben Arous et al., 2021 for learning a single-index model with a known link function via online SGD beyond the Gaussian input setting. The authors introduce a condition called Local Polynomial Growth (LPG) and establish the convergence rate of online SGD under LPG for spherically symmetric input distributions and certain perturbations from the isotropic Gaussian distribution.

**Strengths:**

* The paper has an excellent presentation with sufficient intuition given for different regimes of SGD convergence.
* Learning single-index models for distributions without spherical symmetry is an interesting and significant open problem.
* The calculations performed for spherically symmetric distributions can serve as a useful example of how to perform a harmonic analysis for such distributions in the context of learning with online SGD.

**Weaknesses:**

* My main concern is that LPG might not be the right definition to analyze this problem with non-Gaussian distributions, due to the following reasons:
    * For spherically symmetric distributions, I'm not sure if LPG is required. Specifically, I believe the *information exponent* framework of Ben Arous et al., 2021, which only depends on spherical symmetry and Taylor expansion of the population loss, must be able to handle such distributions, and the contributions in this section might be interpreted as calculating the information exponent for non-Gaussian examples.

    * For distributions without spherical symmetry, it appears that LPG is not the suitable definition. Particularly, as the authors have pointed out, the validity of Assumption 14 for many distributions of interest is not clear, and the probability that SGD has a successful initialization is of order $O(1/d^2)$, making it inapplicable in high-dimensional settings. Moreover, there appears to be no direct connection between the Stein's-method-based argument and LPG.

**Questions:**

* I believe it would be helpful to have a discussion on the connection between the calculations of Section 3 and the framework of Ben Arous et al., 2021. In particular, is showing LPG strictly required for establishing SGD convergence for spherically symmetric distributions, or is it possible to use the Taylor expansion of Ben Arous et al., 2021 to arrive at the same results?

* In line 161, it is mentioned that "This probability can be lowered by any constant factor ...". Is this not the success probability? If that is the case, is this sentence meant to say the probability can be increased?

* In the same sentence, how can one measure the alignment of different initializations with $\theta^*$ in order to maximize such alignment, without having knowledge of $\theta^*$?

**Limitations:**

The authors have mostly adequately discussed the limitations of the work, and additional details are provided above.

---

> ### Author Rebuttal · Authors · 2023-08-09
>
> Thank you for your response and questions, hopefully they will help improve greatly the paper. We give a precise answer to your concerns below:
>
>  - *Necessity of LPG.* Thanks for raising this question that allows us to clarify an important point. In *the Gaussian case*, the definition in the information exponent as done in Ben Arous et al., is sufficient alone as outside the equator $m \neq 0$, **there is no local minima of the population loss**,  as the loss is strictly monotonic for $m > 0$. However, let us put emphasis on the fact that this is not the case in our setup: higher order Gegenbauer polynomials can introduce local minima into the loss landscape, which complicates the definition of information exponent and requires us to control decay of higher order terms as in Proposition 7.
>
> - *No spherical symmetry.* For the first point on the success of initialization, we agree that this is a limitation and we have complemented the result with a discussion indicating that we do not know whether this polynomial number is only a technical limitation or not. Yet, on the second point, we beg to differ. Let us emphasize that the aim of the section 4.2 is to provide the basis of a perturbative setup where the LPG is realised. More precisely, this says that the perturbation of the Gaussian should occur at scale $1/\sqrt{d}$. In that direction, the Stein's method gives nearly (except in small regions with sparse predictors) a positive result for product measures: that is to say that, in high-dimension, for product measure, Stein's method and LPG are well connected! We will make this important fact clearer.
>
> - *Taylor expansion and information exponent.* As said before, the Taylor expansion of Ben Arous et al. 2021 cannot be applied simply because, in the rotationally invariant case, the expansion will include negative coefficients and therefore spurious minima.
>
> - *Initialization.* For line 161, we wanted to say that if we initialized randomly a batch of different $\theta_0$, then the probability of one being well initialized (i.e. success) will be lowered significantly.  You are right that it is *a priori* impossible to measure the alignment without the knowledge of $\theta_*$. What can be done in practice is to run in parallel all the dynamics with different initializations and to take, at the end of training, the one with smallest empirical error (wrt the samples used for SGD). We will clarify this, thanks for raising this problem.

---

> > ### Comment · Reviewer_PMRM · 2023-08-12
> >
> > Thank you for your detailed response. Given the handling of additional stationary points outside the equator, I now have more confidence that the LPG condition provides a framework that is meaningfully stronger than the analysis of Ben Arous et al., 2021 in the spherically symmetric setting, and I have changed my score accordingly. However, to my understanding, the convergence analysis under LPG can be reduced to the convergence analysis of Ben Arous et al., 2021, since the mechanism to avoid the undesirable stationary points is by relying on initialization outside of a band of order $O(1/\sqrt{d})$. Thus, one could modify Assumption A of Ben Arous et al., 2021 to hold outside this band rather than the entire hemisphere and recover the same results. If my interpretation is correct, then it is perhaps worth highlighting that the most interesting contributions are in Section 4.1 where the validity of LPG for rotationally invariant distributions is established under certain technical conditions.
> >
> > As a suggestion, since the authors have decided to cite papers that are in non-identifiable settings, there are other works recently published in similar venues which the authors may like to consider citing for a more extensive list of related works:
> >
> > * Jimmy Ba, Murat A Erdogdu, Taiji Suzuki, Zhichao Wang, Denny Wu, and Greg Yang, “High-dimensional Asymptotics of Feature Learning: How One Gradient Step Improves the Representation”, NeurIPS 2022.
> >
> > * Alireza Mousavi-Hosseini, Sejun Park, Manuela Girotti, Ioannis Mitliagkas, and Murat A Erdogdu, “Neural networks efficiently learn low-dimensional representations with SGD”, ICLR 2023.
> >
> > * Matus Telgarsky, "Feature selection and low test error in shallow low-rotation ReLU networks", ICLR 2023.

---

> > > ### Author Response · Authors · 2023-08-14
> > >
> > > Thanks for your message.
> > >
> > > - Indeed, the main contribution is to understand that typically undesirable stationary points lie "only" in a band of scale $1/\sqrt{d}$ and to prove it through harmonic analysis techniques.  The fact that, if initialized outside this band, SGD typically does not come back into this bad region, is an adaptation from the result proven by Ben Arous et al. This is the main idea of the paper, even if technically, this requires to be extra-careful for some steps. We will be more clear about this.
> > >
> > > -  Thanks for the extra literature.

---

### Official Review · Reviewer_tgdy · 2023-07-05

**Soundness:** 3 good
**Presentation:** 3 good
**Contribution:** 2 fair
**Rating:** 6
**Confidence:** 3

**Summary:**

The paper deals with the study of online SGD in single-index models of the type $f(\mathbf x)=\phi(\boldsymbol\theta^\intercal\mathbf x)$ for a generic non-linear scalar link function $\phi$. The work builds on previous results of Ben Arous et al and aims at going beyond the Gaussian-data assumptions therein. A theorem extending the Gaussian result is given where a crucial condition on the population loss, namely the local polynomial growth, is identified. Finally, the theoretical results are applied to the spherically symmetric setting and to specific non-spherical symmetric settings.

**Strengths:**

The paper is an interesting attempt to go beyond the Gaussian data assumption in the study of online SGD in single-index models and provides a criterion for preserving the scenario observed in the Gaussian setting, characterizing therefore its robustness. If on one hand, the theoretical analysis seems to become extremely complex going beyond the Gaussian realm, it is nevertheless essential to better understand the SGD dynamics on real data. The paper is quite clear although the analysis in Section 4 is more cumbersome (but I have the feeling that this is unavoidable).

**Weaknesses:**

I have the impression that a weakness of the paper lies in the fact that the main information on the loss landscape in the general setting is concentrated on a property, the local polynomial growth condition, which is in general not straightforwardly analyzable. If, on one hand, the typical case discussed in Section 4.1 exemplifies the application of the theorem, on the other hand, it also seems to show that verifying the LPG property is not trivial and requires a setting in which a proper basis (in the case, the Gegenbauer polynomials) is available. The study in Section 4.2, mostly instead relies on a Gaussian reference measure, which, as the authors themselves stress, is also a limitation. I wonder therefore if the theorem is *practically* useful beyond a small set of adequate measures that actually satisfy the LPG property (in this sense the title sounds partially misleading a

**Questions:**

As discussed in the *Weaknesses* section, it appears that verifying the LPG property requires a case-by-case, very nontrivial analysis of the distribution involved. Could the author comment on this point? Do they expect that a general (even heuristic) strategy is feasible to expect to be within the hypothesis of Theorem 3 in the case of a general measure? Finally, with respect to future developments, do the authors expect the LPG condition to be necessary for Theorem 3 to hold?

As a final comment, the authors might consider citing in their work the very classical contribution
> David Saad and Sara A. Solla, Phys. Rev. E 52, 4225 (1995)

where the dynamic of $m$ was studied within the framework of statistical mechanics.

Typos and other comments.
* The authors refer to the goal function as a *Sparse high-dimensional function* but there is no sparsity assumption on, e.g., $\theta$ and, if referring to the fact that the goal is to learn a single direction, the terminology appears slightly misleading.
* In line 70, $\theta^*$ in the definition of $\mathcal H$ should be $\theta$.
* In line 76 $\|f\|^p_{L_\mu^p}$ should be $\|f\|^p_{L_\nu^p}$.
* In lines 248-249, two subsequent "Indeed" appear.
* In line 260 "Wassertein" in place of "Wasserstein".


**Limitations:**

See the *Weaknesses* section.

---

> ### Author Rebuttal · Authors · 2023-08-09
>
> Thank you for your response, comments and paper suggestion that will help improve the paper. We answer precisely to the questions below:
>
> We agree that the LPG condition requires an input distribution for which one can at least approximate the loss landscape in terms of the correlation. Surely, it is very hard, if not impossible, to verify such a property *a priori*. However, we want to argue that the paper aims at understanding what intrinsinc features of the Gaussian distribution helps to analyze a loss landscape: in this paper we give such necesarry conditions. Yet, we still believe it is a useful way of addressing multiple settings, since it clarifies the martingale and drift argument for weak and strong recovery in a way we can apply to multiple settings.
>
> Obviously, we do not expect that LPG hold for every measure, as there as many settings, e.g. described in [Yehudai, Ohad, 2020, Wu, 2022] where the single-index loss landscape is totally *benign* in terms of sample complexity. Yet we believe that the Gaussian distribution offers a rich picture for this problem, and furthermore has a sense of universality as being central in probability and learning. In this direction, the aim of this work was to describe setups where there is this flavor of *universality* as the high-dimensional loss landscape resembles the one canonically given by the Gaussian distribution. Finally, let us mention that the LPG condition is, in some sense necessary if we want to go beyond the Gaussian setting where we know that the dynamics spends most of its time to correlate weakly with the signal.

---

### Official Review · Reviewer_VAjP · 2023-07-06

**Soundness:** 3 good
**Presentation:** 3 good
**Contribution:** 2 fair
**Rating:** 6
**Confidence:** 4

**Summary:**

The main contributions of the authors are as follows:

	(a) Rather than assuming the features are i.i.d. Gaussian, the authors identify a condition called Local Polynomial Growth (LPG, see Def. 2) on the population loss, under which the analysis of Ben Arous et. al. still goes through without any change in their conclusions (see Theorem 3 and Theorem 4 in the paper). Roughly speaking this condition posits a lower bound on the directional derivative of the population loss along the hidden direction at a given point $\theta$ in terms of the overlap between $\theta$ and $\theta_\star$.

	(b) In Proposition 7, the authors then verify that the LPG condition holds when the features are sampled from a possibly non-Gaussian, rotationally invariant distribution (for e.g., the uniform distribution on the sphere).

	(c) In Proposition 15, the authors show that if the 2D projected Wasserstein distance (see eq. 18) between the feature distribution and the standard Gaussian scales like $1/sqrt(d)$ ($d$ is the feature dimension), then for population losses whose Gaussian information exponent is at most $2$ satisfy the LPG condition.

	(d) In Proposition 17, the authors make some partial progress in verifying a relaxed form of the LPG condition for i.i.d. non-Gaussian feature vectors. The basic idea here is that since the LPG condition only depends on a two-dimensional projection of the feature vectors, because of the central limit theorem, the LPG condition should hold. The authors conjecture that SGD path avoids localized vectors, and based on this conjecture, sketch an informal argument that would show that the conclusions of Ben Arous et. al. should also extend to feature vectors with i.i.d. coordinates.

**Strengths:**

I found the paper to be well-written and the proofs (as far as I checked them) seem correct to me. The SGD analysis of Ben Arous et. al. does not assume that the features are i.i.d. Gaussian. specifically, but assume a condition on the population loss (see Eq. 1.1 in the ``Online stochastic gradient descent on non-convex losses from high-dimensional inference'', JMLR 2021),  and analyze SGD dynamics under this condition. However, in most of their examples, the condition can only be verified under the i.i.d. Gaussian feature assumption. The condition proposed by the authors is weaker than the condition in the work of Ben Arous et. al. and can be verified in examples beyond those considered in the work of Ben Arous et. al. This is a key strength of this paper.


**Weaknesses:**

The key weakness of this paper seems to be that most of the examples discussed by the authors seem to be small perturbations of Gaussian sensing vectors.

At the same time, the abstract and the introduction seem to overclaim the generality of the results obtained in the paper. For instance, in the abstract, the authors write ``our main results establish that Stochastic Gradient Descent can efficiently recover the unknown direction in the high dimensional regime, under mild assumptions that significantly extend Yehudai and Ohad (2020) and Wu (2022)''. However, the papers cited by the authors appear to make weaker assumptions on the feature vectors (compared to rotationally invariant features or i.i.d. features), but stronger assumptions on the non-linear link functions, so it appears that the results of the authors are not stronger or weaker - rather complementary.

A second weakness of the paper is that the analysis of the authors for i.i.d. feature vectors seems incomplete. It does seem quite plausible that the conclusions of Ben Arous et al. should extend to this case. However, the claims of the authors are conditional on a conjecture (see Conjecture 35 in the supplement). Furthermore, the fact that their results are conditional on this unproven conjecture is not transparently indicated in the main paper, where the authors simply write ``While the LPG property does not directly apply in this setting, in Appendix B.4 we outline an energy-barrier argument that demonstrates that the single-index model 324 can still be efficiently solved using gradient-based methods'', without mentioning that the result relies on a conjecture they haven't proven.

**Questions:**

It would be great if the authors could clarify the following:

	1) In Theorem 4, is the scaling of the iteration bound with the desired recovery accuracy $\epsilon$ expected to be optimal? This seems important since it would clarify if SGD achieves the optimal statistical rate of convergence. In addition, the result stated in Theorem 4 seems inconsistent with the result obtained in the proof (Page 16 in the supplement).

	2) In Prop 7, the authors consider the setting where the features are rotationally invariant. In this situation, I wonder if it might be possible to recover this result from the results of Ben Arous et. al. since it seems to me that the main assumption of Ben Arous et. al. holds in this case (that is, the population loss at a point $\theta$ is only a function of the dot product between $\theta$ and the hidden direction).

	3) A relevant reference to cite is the work ``One-bit compressed sensing with non-Gaussian measurements'' by Albert Ai, Alex Lapanowski, Yaniv Plan, and, Roman Vershynin. This work also uses a central limit theorem-type argument to go beyond i.i.d. Gaussian measurements to i.i.d. non-Gaussian measurements, as in Section 4.2 of this paper.

I have the following minor comments:

	(1) I think the discussion of the work of Ben Arous et. al. can be improved to clarify that their analysis of SGD doesn't use Gaussianity per se, but is under a general condition, which can be verified for i.i.d. Gaussian features and rotationally invariant features.

	(2) In the introduction, lines 55-57, I didn't quite understand what the authors mean by "stability".

	(3) The discussion in lines 162-164 seems a bit inappropriate: the authors propose drawing many random initializations and keeping the one that has the maximum correlation with the hidden direction - but the hidden direction is unknown.

	(4) Below Eq: 18: Has the notation $Gr(2,d)$ been defined?

---

> ### Author Rebuttal · Authors · 2023-08-09
>
> Thank you for your response, we answer questions below:
>
> **General answer**
>
> - *Tempering the overclaim.* Regarding Yehudai and Ohad (2020) and Wu (2022)', both contain theorems specifically demonstrating strong recovery under specific family of functions and spherically rotationally invariant distributions (these are Theorem 6.4 and Theorem 3.3, respectively).  It is true that Theorem 5.3 in the former paper considers a setting we don't extend but is complementary to our own, we will update the paper to reflect this fact. However, we want to be clear that we did not cover only the "small perturbation of Gaussian case": indeed the Section 4.1 is devoted to a setup that can be far from Gaussian, while preserving its spherical symmetry.
>
> - *Conjecture in the case of product measure.* We want to be clear that the only case that remains conjectural is at the end of the paper when the product measure is introduced. We see this paragraph (appearing only page 9!) as a nice perspective that carries a *universality* flavor, but as said by the reviewer, we never claim to have fully solved it: we do not see this as a weakness but rather as a nice perspective that naturally arises with our work. We will make this more clear in the revised manuscript.
>
>
> **Answer to specific questions.** Thanks for the comments and the proposed (relevant) paper, we will add these in a later draft.
>
> 1) *Epsilon dependency.* Note that all that matters in this strong recovery phase is that this dependency *is independent of the dimension*: this is the critical point of this theorem. The precise dependency on *$\varepsilon$* is not very important in that sense. In fact, all the time spent is in the first phase described by Theorem 3. To be clear about this fact, we decided, as it is done implicitly in [Ben Arous et al]'s work to reschedule the step-size. Thanks to point out this fact, this will be clearer in the updated version.
>
> 2) *Rotation invariance does not suffice.* This is one of the principal conclusion of our study: the result does not immediately follow from Ben Arous, as for spherically symmetric inputs that are non-Gaussian, the loss is not globally monotonic.  Indeed, controlling Gegenbauer polynomials to determine how large the initial correlation must be to land in the monotonic part of the loss landscape is the bulk of the proof: in the Gaussian case higher order Hermite terms cannot possibly hurt you (they can only make the loss decrease faster), whereas in our setting we need to carefully control the higher order terms in order to avoid local minima.
>
> 3) *Initialization.* Thank you for pointing out this fact! It is true we cannot directly measure the correlation between $\theta$ and $\theta^*$.  However, for large $d$ and constant values of $s$, the LPG condition implies that once we've reached weak recovery and e.g. $m \geq 1/2$, the loss landscape is monotonic.  This implies we can, after training several possible $\theta$ in parallel, test which one has the highest correlation with $\theta^*$ by simply looking at which one minimizes the empirical loss over a batch of fresh samples. We'll make this important fact clearer.
>
> 4) *Grassmannian.* $Gr(2,d)$ is a notation for the Grassmannian of two-dimensional subspaces (the manifold of all the two-dimensional linear spaces), but we didn't include this definition explicitly and will update in a later draft.

---

> > ### Comment · Reviewer_VAjP · 2023-08-17
> > **Acknowledgement**
> >
> > I have read the authors' response and I am grateful to them for their engagement. My overall assessment of the paper is unchanged and continues to lean towards acceptance.

---

### Official Review · Reviewer_kUyA · 2023-07-07

**Soundness:** 3 good
**Presentation:** 2 fair
**Contribution:** 3 good
**Rating:** 5
**Confidence:** 4

**Summary:**

This paper studied the online SGD of learning a single-index teacher model with the link function as the student model. This work generalized the results of [Ben Arous et al. 2021], where the information exponent of the link function governs the sample complexity, to the non-Gaussian dataset, where the author raised a local polynomial growth (LPG) to describe the loss landscape.  This LPG prevents the loss landscape to have bad local minima outside a band of the equator. The author provided two cases of loss landscape with LPG: (i) in the spherically symmetric case, similar behavior as the Gaussian case is exhibited; (ii) in the non-symmetric case, more assumptions on link function and feature direction $\theta^*$ are needed and the deviation between the non-Gaussian and Gaussian distribution in certain metric characterizes the LPG property.

**Strengths:**

Overall, the presentation of this paper is clear, and the motivation of the paper is strong. It is important to generalize online SGD with non-Gaussian data distributions and extend the information exponent of the link function in this general setting. The results obtained by the authors are as sharp as can be expected given the problem under consideration. Although the model studied is simple, the insights derived are non-trivial and the technical approach is original. Especially, in general cases, this paper introduces a two-dim Wasserstein distance to measure the deviation from the Gaussian reference and Stein method for product measures, which are interesting directions for future work.

**Weaknesses:**

1. Simulations like [Ben Arous et al. 2021] did should be included in the paper. Especially certain non-Gaussian distributions either spherically symmetric or not, should be presented in the simulations to see how information exponent and LPG control the weak recovery and training dynamic.
2. More examples of non-Gaussian distribution could be mentioned in Section 4. Like Example 8 and 9, there should be some examples for non-spherically symmetric cases in Section 4.2.
3. Unlike [Ben Arous et al. 2021], there is no lower bound showing the smallest sample complexity for efficient online SGD using LPG in non-Gaussian cases. In the Gaussian case, $n\ge d^{s-1}$ is nearly tight for online SGD where $s$ is the information exponent.

**Questions:**

1. [Ba et al. 2022] is another paper addressing a semi-parametric problem when the link function is unknown with certain GD training. [Arnaboldi et al. 2023] also studied the SGD training of two-layer neural networks in high-dimensional settings.
2. Can we consider noisy labels in (1)?
3. Line 106: information exponent $s$ should be explicitly defined here.
4. [Tan and Vershynin 2019] considered uniform sphere data distribution, although the link function is only quadratic in this phase retrieval case. The SGD dynamics for the non-Gaussian phase retrieval model may be analyzed in the literature.
5. Line 154 double "that"
6. Between lines 189 and 190, what is distribution $\rho$ when we define measure $\nu$?
7. In Proposition 7, is there some relation between $s^*$ information exponent $s$ as $d\to\infty$? There is a formula describing the relation between Hermite coefficients and spherical harmonic coefficients. Hence, asymptotically, will the results of Example 8 return to the Gaussian case or not?
8. How about the problem of model misspecification, where the link function of the target is not known?

===================================================================================================
- Ba, J., Erdogdu, M.A., Suzuki, T., Wang, Z., Wu, D. and Yang, G., 2022. High-dimensional asymptotics of feature learning: How one gradient step improves the representation. Advances in Neural Information Processing Systems, 35, pp.37932-37946.
- Arnaboldi, L., Stephan, L., Krzakala, F. and Loureiro, B., 2023. From high-dimensional & mean-field dynamics to dimensionless ODEs: A unifying approach to SGD in two-layers networks. arXiv preprint arXiv:2302.05882.
- Tan, Y.S. and Vershynin, R., 2019. Online stochastic gradient descent with arbitrary initialization solves non-smooth, non-convex phase retrieval. arXiv preprint arXiv:1910.12837.

**Limitations:**

I think the authors have adequately addressed the potential negative social impact of their work.

---

> ### Author Rebuttal · Authors · 2023-08-09
>
> Thank you for your review, our response follows:
>
> **Main response**
>
> - *Simulations.* We've included simulations of learning single index models that verify our theory under rotationally invariant and non-Gaussian input distributions in a new draft of the paper.  Specifically, we see that inputs uniform on the sphere only reach strong recovery when the random initialization exceeds the last zeros of the associated Gegenbauer polynomial's largest root, and iid hypercube inputs can reliably learn functions with s = 2 but not s = 3.
>
> - *Examples of non-spherically symmetric.* Typical examples are in fact given in the paragraph between lines 294 and 303. We will state this more precisely as we have done in Examples 8 and 9. Going further, let us mention that studying thoroughly this family of perturbed Gaussian distribution is a relevant (but hard!) question on its own and we consider this study for future investigations.
>
> - *Lower bound.* Generally speaking, a tight lower bound for learning *general* spherical distributions is a desirable result, but not at all as simple as the Gaussian case.  The theory provides a necessary setting for the LPG condition, but unlike the Gaussian case, there can be advantage to choosing a different link function for the student model than the teacher model.  In other words, the range where the loss landscape will be increasing as a function of the correlation $m$ will depend very carefully on the orthogonal polynomials of the radial distribution. That being said, for *specific* problems, as the uniform distribution over the sphere or (obviously!) the Gaussian case, it is relevant to point out it is possible to derive such lower-bounds. We'll add a commentary on this fact in the revised manuscript.
>
> **Answer to specific questions.**
>
> First, thank you for the additional papers and the typos, we'll add/correct them and this will help improving the overall quality of the manuscript.
>
> - *Noisy labels.* Yes, subgaussian noisy labels shouldn't make any impact on the analysis since this would simply add an extra noise term in the growth of the correlation $m_t$ that is negligible in the step size regime we are considering. We'll add a comment on this fact.
>
> - *Phase retrieval.* The paper [Tan and Vershynin 2019] in indeed *included* in our setting. This corresponds to $s = 2$, and the second degree Gegenbauer polynomial being a simple parabola, as long as $|x|$ satisfies the decay conditions of Proposition 7, this fits into our framework.
>
> - *Asymptotic case where $d \to \infty$.* Asymptotically, yes the Gegenbauer polynomials approach Hermite polynomials (under appropriate scaling by the dimension) as $d \to \infty$, so in this limit this returns to the Gaussian case and $s^*$ and $s$ will match.
>
> - *Unknown link function*. Even in the Gaussian case, the story that comes out if the link function is unknown is totally different, and may be subtle. We refer to articles already mentioned in the main text lines 45-46 to show that the way the unknown function is simultaneously learn changes totally the picture. It could be interesting to extend the present analysis to such a framework, but this is obviously not the purpose of the present work.

---

> > ### Comment · Reviewer_kUyA · 2023-08-19
> > **Thank you for the rebuttal**
> >
> > Thank you for your rebuttal and significant effort. The authors provided reasonable answers to my questions and comments. I think it would be more convincing if they could attach their experiment results, especially the non-symmetric examples, in the global rebuttal before. Here are a few replies and comments.
> >
> > 1. It would be great to, in the main text, emphasize the necessity of LPG and the possible local minimum of the loss landscape in non-Gaussian distribution cases.
> >
> > 2. As for the tight lower bound in the main response, I agree that we can derive tight lower bounds for uniform distribution over the sphere case and it is harder for general spherical distributions. Are you going to verify this for the uniform spherical distribution case in the revision? And do you have a conjectured condition on spherical radius distribution we may need to obtain a tight lower bound in the general case?
> >
> > 3. Based on Reviewer VAjP, Conjecture 35 in the appendix should be mentioned in the main text, otherwise the readers may misunderstand what you have proved. You can also provide some simulations to empirically verify this conjecture is right in certain cases.

---

> > > ### Author Response · Authors · 2023-08-21
> > >
> > > Thank you for you comments.
> > >
> > > **Experiments.** We are very sorry for this but as far as we understood, we cannot send a pdf with additional experiments any more. To try to depict this a bit: for the non-symmetric example, we took the tensorized distribution on the hypercube $\mathrm{Unif}([0,1])^{\otimes d}$ (with $d = 50$) and compared the success of the correlation escaping mediocrity with the Gaussian case with two different link functions $f_2^* = \frac{1}{2} \left(h_2 - h_3 - h_4 + h_5 \right)$ (this corresponds to $s = 2$) and $f_3^* = h_3$ (this corresponds to $s = 3$). For the second degre case, escape appeared frequently both for Gaussian and uniform over the hypercube , but for the third degree case (outside of the theoretical scope of the paper), the case with hypercube distribution failed frequently to learn validating, in some sense, that $s= 2$ corresponds to a limit case.
> > >
> > > **Possible local minima.** We will definitely clarify this point: it is one of the major difference with the Gaussian case and should be emphasize. We'll add a commentary on this.
> > >
> > > **Lower bound.** Deriving a lower bound is always a demanding exercise (even in the spherical case), and we cannot promise at this point that we'll do this for the revisions. We will add a commentary on how this could be done (and may be addressed in the general case).
> > >
> > > **Conjecture.** Sure, we have put emphasis on Conjecture 35 in the revised version and put this in the main text of the paper.

---

### Comment · Area_Chair_MtaE · 2023-08-18
**Thank you for the rebuttal**

Dear authors,

thank you for providing a rebuttal. Some of the reviewers have already replied, so this is just to let you know that I am in contact with the remaining ones as well.

Best,
Your AC

---

> ### Author Response · Authors · 2023-08-18
>
> Thank you for this message,
> we are at the reviewer's disposal for any additional precisions or comments.
> Best,
> The authors.

---

### Decision · Program_Chairs · 2023-09-21

**Decision:**

Accept (poster)

**Comment:**

This paper studies the behaviour of gradient descent for learning a single-index model. Recent work by Ben Arous et al. has provided a precise description of online SGD, assuming Gaussian data. The key contribution of this work is to go beyond Gaussian data, and the authors identify an assumption (called Local Polynomial Growth) under which the existing analysis and the corresponding conclusions go through.

The authors have addressed the concerns of the reviewers via a thoughtful rebuttal, and the consensus is towards accepting the paper. After my own reading of the reviews, rebuttal and paper, I agree with this view: removing Gaussian assumptions is an important problem and, even if the LPG may be hard to verify, this paper still does a step in the right direction providing interesting and non-trivial results. I thus recommend acceptance.

I warmly encourage the authors to incorporate the rebuttal in the final version. I would also like to see a comparison with the classical work "One-bit compressed sensing with non-Gaussian measurements" by Ai et al. (https://arxiv.org/abs/1208.6279). In fact, the model considered in that work can also be regarded as a single index model with non-Gaussian measurements.